# Epidemiological and ecological determinants of Zika virus transmission in an urban setting

José Lourenço[1]*, Maricelia Maia de Lima[2], Nuno Rodrigues Faria[1], Andrew Walker[1], Moritz UG Kraemer[1], Christian Julian Villabona-Arenas[3], Ben Lambert[1], Erenilde Marques de Cerqueira[4], Oliver G Pybus[1], Luiz CJ Alcantara[2], Mario Recker[5]

[1]Department of Zoology, University of Oxford, Oxford, United Kingdom; [2]Laboratory of Haematology, Genetics and Computational Biology, FIOCRUZ, SalvadorBahia, Brazil; [3]Institut de Recherche pour le Développement, UMI 233, INSERM U1175 and Institut de Biologie Computationnelle, LIRMM, Université de Montpellier, Montpellier, France; [4]Centre of PostGraduation in Collective Health, Department of Health, Universidade Estadual de Feira de Santana, Feira de SantanaBahia, Brazil; [5]Centre for Mathematics and the Environment, University of Exeter, Penryn, United Kingdom

*For correspondence:
jose.lourenco@zoo.ox.ac.uk

Competing interests: The authors declare that no competing interests exist.

**Abstract** The Zika virus has emerged as a global public health concern. Its rapid geographic expansion is attributed to the success of *Aedes* mosquito vectors, but local epidemiological drivers are still poorly understood. Feira de Santana played a pivotal role in the Chikungunya epidemic in Brazil and was one of the first urban centres to report Zika infections. Using a climate-driven transmission model and notified Zika case data, we show that a low observation rate and high vectorial capacity translated into a significant attack rate during the 2015 outbreak, with a subsequent decline in 2016 and fade-out in 2017 due to herd-immunity. We find a potential Zika-related, low risk for microcephaly per pregnancy, but with significant public health impact given high attack rates. The balance between the loss of herd-immunity and viral re-importation will dictate future transmission potential of Zika in this urban setting.
DOI: https://doi.org/10.7554/eLife.29820.001

## Introduction

The first cases of Zika virus (ZIKV) in Brazil were concurrently reported in March 2015 in Camaçari city in the state of Bahia (*Campos et al., 2015*) and in Natal, the state capital city of Rio Grande do Norte (*Zanluca et al., 2015*). During that year, the epidemic in Camaçari quickly spread to other municipalities of the Bahia state, including the capital city of Salvador, which together accounted for over 90% of all notified Zika cases in Brazil in 2015 (*Faria et al., 2016a*). During this period, many local Bahia health services were overwhelmed by an ongoing Chikungunya virus (CHIKV, East Central South African genotype) epidemic, that was first introduced in 2014 in the city of Feira de Santana (FSA) (*Nunes et al., 2015*; *Faria et al., 2016b*). The role of FSA in the establishment and subsequent spread of CHIKV highlights the importance of its socio-demographic and climatic setting, which may well be representative for the transmission dynamics of arboviral diseases in the context of many other urban centres in Brazil and around the world.

On the 1$^{st}$ February 2015 the first ZIKV cases were reported in FSA, followed by a large epidemic that continued into 2016. The rise in ZIKV incidence in FSA coincided temporally with an increase in

**eLife digest** Mosquitoes can transmit viruses that cause Zika, dengue and several other tropical diseases that affect humans. Zika virus usually causes mild symptoms, but it is thought that infection during pregnancy can lead to brain abnormalities, including microcephaly, where babies are born with an abnormally small head. Recent studies have shed light on how the Zika virus spread from Africa to reach South America, the Caribbean and North America. However, much less is known about the ecological factors that contribute to the spread of the virus within towns, cities and other local areas.

In 2015, Brazil was struck by an outbreak of the Zika virus that led to an international public health emergency. Lourenço et al. used a mathematical model to explore the local conditions within Feira de Santana (a major urban center in Brazil) that contributed to the outbreak. The model took into account numerous factors, including temperature, humidity, rainfall and the mosquito life-cycle, which made it possible to reconstruct the history of the virus over the past three years and to make projections for the next decades.

It revealed that most of the infections occured during 2015, with approximately 65% of the population infected. The incidences of new infections declined in 2016, as increasing numbers of local people had already been exposed to the virus and became immune. Temperature and humidity appeared to have played a critical role in sustaining the mosquito population carrying the Zika virus.

Further analysis suggests that the risk of Zika virus causing microcephaly is very low – only 0.3–0.5% of the pregnant women in Feira de Santana who were infected with Zika gave birth to a baby with the condition. What therefore makes Zika a public health concern is the combination of a low risk with very high infection rates, which can affect a large number of pregnancies.

This study will help researchers and policy makers to predict how the Zika virus will behave in the coming years. It also highlights the limitations and successes of the current system of surveillance. Moreover, it will help to identify critical time periods in the year when mosquito control strategies should be implemented to limit the spread of this virus. In future, this could help shape new local strategies to control Zika virus, dengue and other diseases carried by mosquitoes.
DOI: https://doi.org/10.7554/eLife.29820.002

cases of Guillain-Barré syndrome (GBS) and microcephaly (*Faria et al., 2016a*), with an unprecedented total of 21 confirmed cases of microcephaly in FSA between January 2015 and May 2017. There is wide statistical support for a causal link between ZIKV and severe manifestations such as microcephaly (*Rubin et al., 2016*; *de Araújo et al., 2016*; *Soares de Araújo et al., 2016*; *Honein et al., 2017*; *Brasil et al., 2016*; *de Oliveira et al., 2017*), and the proposed link in 2015 led to the declaration of the South American epidemic as an international public health emergency by the World Health Organization (WHO) in 2016; the response to which has been limited to vector control initiatives and advice to delay pregnancy in the affected countries (*WHO, 2016b*; *WHO, 2016a*). With few cohort studies published (*Honein et al., 2017*; *Brasil et al., 2016*) and the lack of an established experimental model for ZIKV infection (*Aman and Kashanchi, 2016*; *Dowall et al., 2016*), modelling efforts have taken a central role for advancing our understanding of the virus's epidemiology (*Chowell et al., 2016*; *Ferguson et al., 2016*; *Bogoch et al., 2016*; *Nishiura et al., 2016*; *Zhang et al., 2016*; *Perkins et al., 2016*; *Messina et al., 2016*). In particular, our knowledege on parameters of public health importance, such as the basic reproduction number ($R_0$), the duration of infection (*Ferguson et al., 2016*), attack and reporting rates (*Kucharski et al., 2016*), the risk of sexual transmission (*Maxian et al., 2017*; *Gao et al., 2016*; *Moghadas et al., 2017*) and birth-associated microcephaly (*Bewick et al., 2016*; *Perkins et al., 2016*) has advanced significantly from studies using transmission models. Climate variables are critical for the epidemiological dynamics of Zika and other arboviral diseases, such as dengue (*Lourenço and Recker, 2014*; *Feldstein et al., 2015*; *Kraemer et al., 2015*; *van Panhuis et al., 2015*) and chikungunya (*Poletti et al., 2011*; *Mourya et al., 2004*; *Salje et al., 2016*). Although these have also been previously addressed in mapping and/or modelling studies (e.g. (*Bogoch et al., 2016*; *Zhang et al., 2016*; *Perkins et al., 2016*; *Messina et al., 2016*)), their effects as ecological drivers for the

emergence, transmission and endemic potential of the Zika virus, especially in the context of a well described outbreak, have not yet been addressed in detail.

In this study, focusing on an urban centre of Brazil (Feira de Santana), we explicitly model the mosquito-vector lifecycle under seasonal, weather-driven variations. Using notified case data of both the number of suspected Zika infections and confirmed microcephaly cases, we demonstrate how the combination of high suitability for viral transmission and a low detection rate resulted in an extremely high attack rate during the first epidemic wave in 2015. The rapid accumulation of herd-immunity significantly reduced the number of cases during the following year, when total ZIKV-associated disease was peaking at the level of the country. Projecting forward we find that the demographic loss of herd-immunity together with the frequency of reintroduction will dictate the risk of reemergence and endemic establishment of Zika in Feira de Santana. The conclusions of this study should be transferable to major urban centres of Brazil and elsewhere with similar climatic and demographic settings.

## Methods summary

To model the transmission dynamics of ZIKV infections and estimate relevant epidemiological parameters, we fitted an ento-epidemiological, climate-driven transmission model to ZIKV incidence and climate data of FSA between 2015 and 2017 within a Bayesian framework, similar to our previous work on a dengue outbreak in the Island of Madeira (*Lourenço and Recker, 2014*).

The model is based on ordinary differential equations (ODE) describing the dynamics of viral infections within the human and mosquito populations (*Equations 1-5 and 6-10*, respectively). The human population is assumed to be fully susceptible before the introduction of ZIKV and is kept constant in size throughout the period of observation. After an infectious mosquito bite, individuals first enter an incubation phase, after which they become infectious to a mosquito for a limited period of time. Fully recovered individuals are assumed to retain life-long immunity. We assumed that sexual transmission did not significantly contribute to transmission dynamics and therefore ignored its effects (*Yakob et al., 2016*; *Moghadas et al., 2017*; *Maxian et al., 2017*).

For the dynamics of the vector populations we divided mosquitoes into two life-stages: aquatic and adult females. Adult mosquitoes were further divided into the epidemiologically relevant stages for arboviral transmission: susceptible, incubating and infectious. In contrast to human hosts, mosquitoes remain infectious for life. The ODE model comprised 8 climate-dependent entomological parameters (aquatic to adult transition rate, aquatic mortality rate, adult mortality rate, oviposition rate, incubation period, transmission probability to human, hatching success rate and biting rate), whose dependencies on temperature, rainfall and humidity were derived from other studies (see *Table 1*).

Four parameters (baseline mosquito biting rate, mosquito sex ratio, probability of transmission from human-to-vector and human lifespan) were fixed to their expected mean values, taken from the literature (see *Table 2*). To estimate the remaining parameters, alongside parameter distributions regarding the date of first infection, the human infectious and incubating periods, and the observation rate of notified ZIKV cases, we fitted the ODE model to weekly notified cases of ZIKV in FSA using a Bayesian Markov-chain Monte Carlo (MCMC) approach. The results are presented both in

**Table 1.** Model climate-dependent parameters.

| Notation | Description |
| --- | --- |
| $\epsilon_A^v(t)$ | transition rate from aquatic to adult mosquito life-stages |
| $\mu_A^v(t)$ | mortality rate of aquatic mosquito life-stage |
| $\mu_V^v(t)$ | mortality rate of adult mosquito life-stage |
| $\theta^v(t)$ | (human) intrinsic oviposition rate of adult mosquito life-stage |
| $\gamma^v(t)$ | (vector) extrinsic incubation period of adult mosquito life-stage |
| $\phi^{v \to h}(t)$ | vector-to-human probability of transmission per infectious bite |
| $c^v(t)$ | egg hatching success |
| $a^v(t)$ | adult vector biting rate |

DOI: https://doi.org/10.7554/eLife.29820.003

**Table 2.** Model constant parameters.

| Notation | Value | Description | References |
|---|---|---|---|
| $a^v$ | 0.25 per day | mosquito biting rate | [ 76, 88 ] |
| $f$ | 0.5 | proportion of females (sex ratio) | [ 52, 59 ] |
| $\phi^{h \to v}$ | 0.5 | human-to-vector probability of transmission per infectious bite | – |
| $1/\mu^h$ | 75 years | human mean lifespan | [ 83 ] |

DOI: https://doi.org/10.7554/eLife.29820.004

terms of mean dynamic behaviour of the ODE under the MCMC solutions and posterior distributions of key epidemiological parameters. A full description of the fitting approach and the estimated parameters can be found in the section Materials and methods.

## Results

On the $1^{st}$ February 2015 the first Zika virus (ZIKV) case was reported in Feira de Santana (FSA). Weekly cases remained very low for the following two months, adding up to just 10 notified cases by the end of March that year (*Figure 1A*). A rapid increase in the number of cases was observed in April, coinciding with Micareta, a local carnival-like festival that takes place across the urban centres of Bahia. The epidemic peaked in July 2015, which was followed by a sharp decline in notified cases over the next 1–2 months. This first epidemic wave was followed by a significantly smaller outbreak in 2016, peaking around March, and an even smaller outbreak in 2017 with no discernable epidemic peak.

Confirmed (and monthly aggregated) microcephaly (MC) cases were absent by November 2015, after which a small epidemic was observed with peak counts in January 2016. We found a time lag of 5–6 months (20–24 weeks) between the first reported Zika epidemic wave and the MC peak in case counts. This coincides with previous observations suggesting a link between the development of neurological complications in newborns and ZIKV infection during the second trimester (*Faria et al., 2016a*). We note that our lag may be offset by around 1–4 weeks, however, since the date of MC cases in our dataset represents the date of diagnostic confirmation, which is usually done postpartum.

Overall, the epidemic behaviour in FSA was in sharp contrast with trends observed in notified cases across Brazil (BR) as a whole, for which the second epidemic in 2016 was approximately 6 times larger than the one in 2015 (*Figure 1A*), suggesting the Bahia state as a focus point in the emergence and initial spread of ZIKV in Brazil (*Faria et al., 2016c*; *Faria et al., 2016a*). Nonetheless, a clear temporal synchronization between country level and FSA case counts could be observed.

The age distribution of notified ZIKV cases in FSA suggested a higher proportion of cases between 20 and 50 years of age, but with no discernible differences between the two epidemics (*Figure 1B*, top panel). However, when corrected for the expected number of cases assuming an equal risk of infection per age class, we found the number of cases within this age group to be closer to most other groups (incidence rate ratio, IRR, close to 1, *Figure 1B*, bottom panel). The per capita case counts within the youngest age class (<1 years) appeared higher than expected, with an IRR significantly above 1 and also higher in 2016 (IRR = 4.4, 95% CI [2.8, 7.0]) than in 2015 (IRR = 1.95, 95% CI [1.5, 2.6]), possibly indicating biased reporting and/or health care seeking with increased awareness of the disease. There was also a consistent trend towards reduced IRR in the elderly (>65 years), although with significant uncertainties. Finally, a small increase in IRR could be detected in the 20–34 year olds, which could potentially be a signature of sexual transmission in this age group (*Gao et al., 2016*; *Carlson et al., 2016*; *Foy et al., 2011*; *Turmel et al., 2016*; *Yakob et al., 2016*; *Maxian et al., 2017*; *Moghadas et al., 2017*). At this stage and without more detailed data it was not possible to ascertain whether these findings indicated age-related risk of disease, age-dependent exposure risk or simply notification biases in particular age groups, however.

The spatial distribution of total notified cases for BR highlighted the expected clustering of ZIKV cases within the Bahia state by the end of 2015 as well as the wider geographical range by July 2016 (*Figure 1C*). We speculate that the difference in geographical range could explain the higher number of cases observed during the 2016 epidemic at the country level. This, on the other hand, did

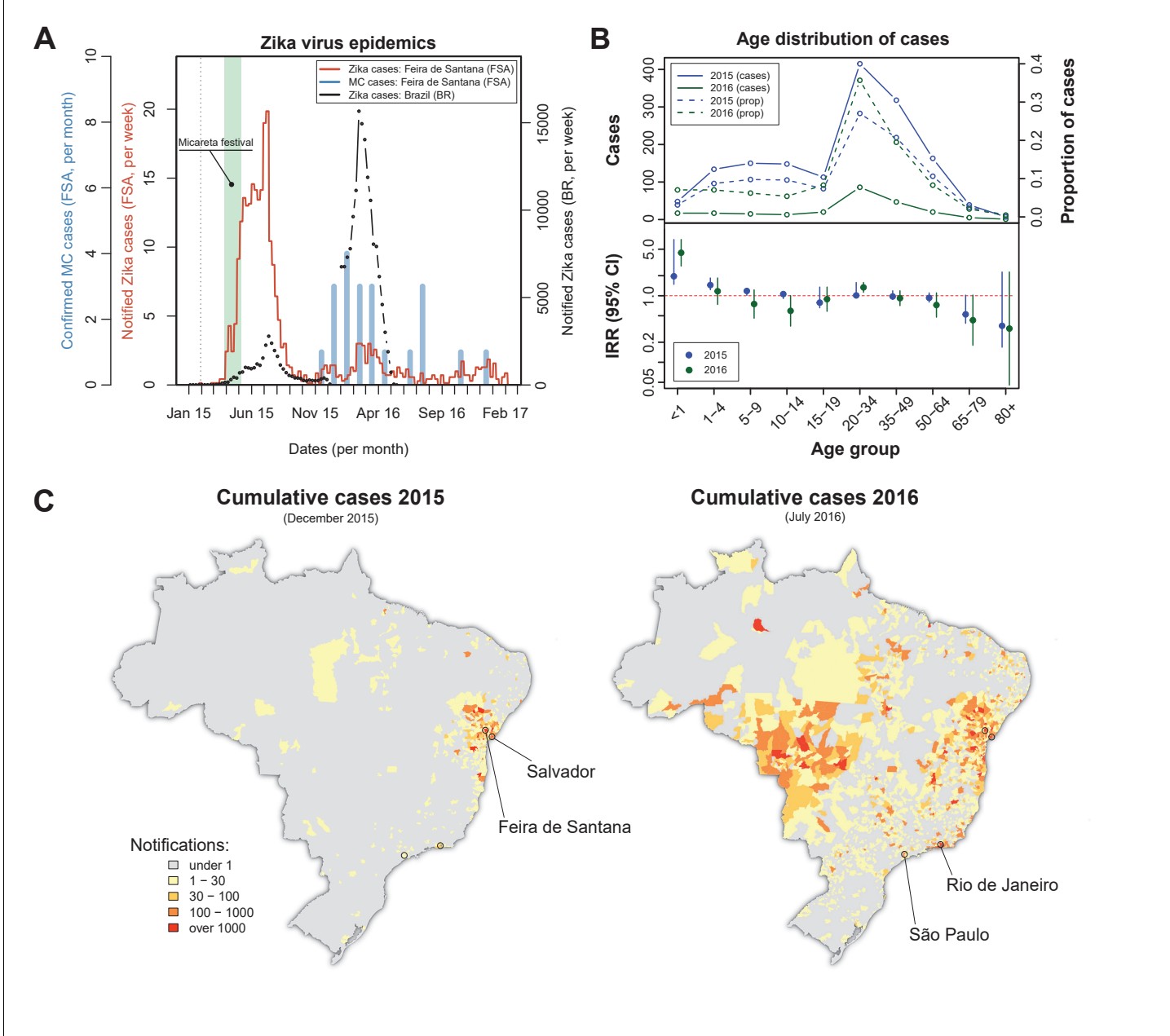

**Figure 1.** Zika virus epidemics in Feira de Santana and Brazil (2015–2016). (**A**) Comparison of weekly notified Zika cases (full red line) with monthly Microcephaly cases (blue bars) in Feira de Santana (FSA), overimposed with total Zika cases at the level of the country (BR, black dotted line). BR data for weeks 50–52 was missing. Green area highlights the time period for the Micareta festival and the dotted grey line the date of first notification. Incidence series is available as Dataset 3 and Microcephaly series as Dataset 4. (**B**) Age distribution and incidence rate ratio (IRR) for the 2015 (blue) and 2016 (green) FSA epidemics (data available as Dataset 2). The top panel shows the number of cases per age (full lines) and the proportion of total cases per age class (dashed lines), which peak at the age range 20–50. The bottom panel shows the age-stratified incidence risk ratio (IRR, plus 95% CI), with the red dotted line indicating $IRR = 1$. (**C**) Spatial distribution of cumulative notified cases in BR at the end of 2015 (left) and mid 2016 (right). Two largest urban centres in the Bahia state (Salvador, Feira de Santana) and at the country level (São Paulo, Rio de Janeiro) are highlighted.

DOI: https://doi.org/10.7554/eLife.29820.005

not explain why the second epidemic in FSA was nearly 7 times smaller than the first and with only sporadic cases in 2017. To answer this question and to obtain robust parameter estimates of ZIKV epidemiological relevance we utilised a dynamic transmission model, which we fitted to notified

case data and local climate variables of FSA within a Bayesian framework (see Materials and methods).

## Climate-driven vectorial capacity

The reliance on *Aedes* mosquitoes for transmission implies that the transmission potential of ZIKV is crucially dependent on temporal trends in the local climate. We therefore investigated daily rainfall,

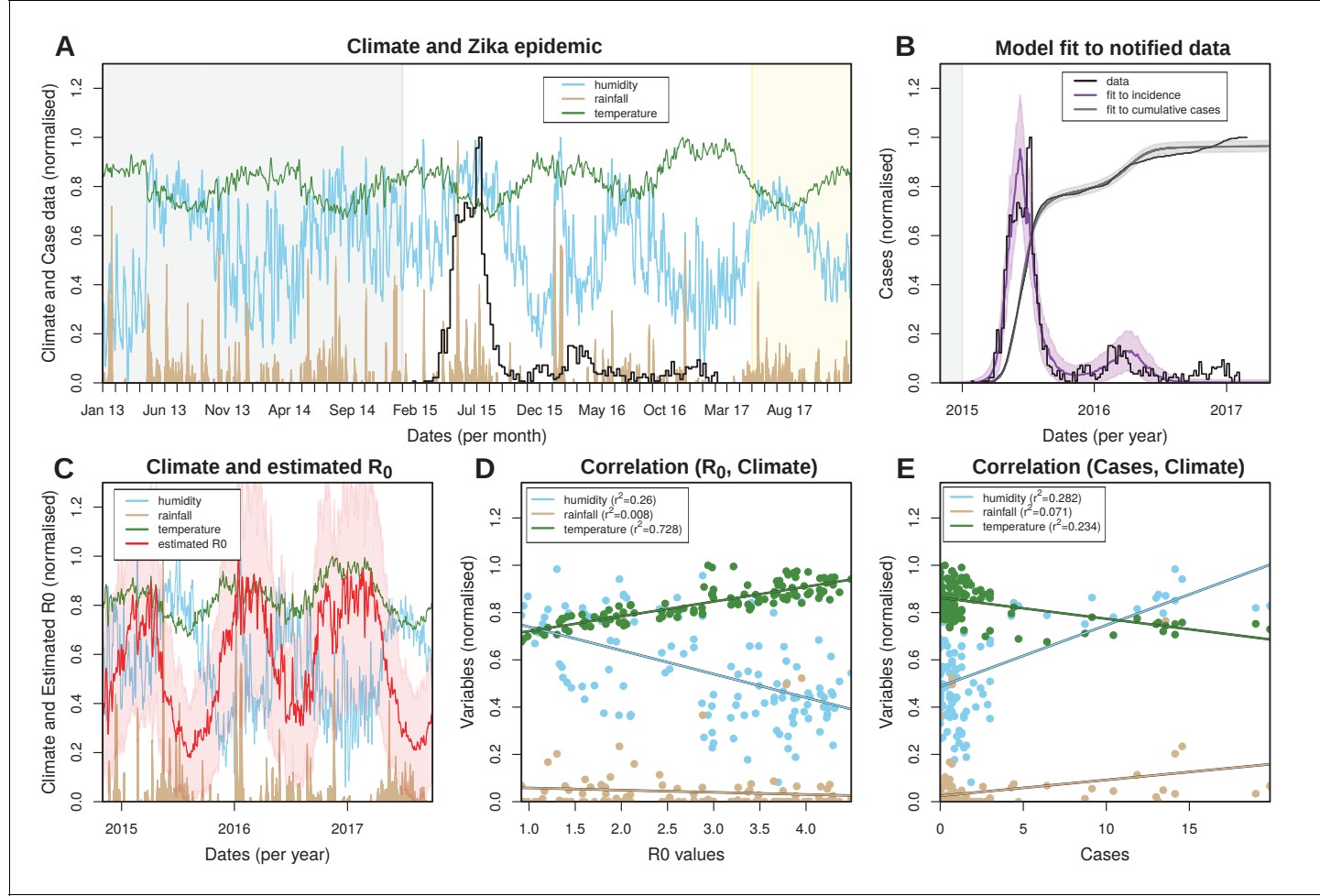

**Figure 2.** Eco-epidemiological factors and model fit to notified cases. (A) Zika case data (black) and daily climatic series for rainfall (gold), humidity (blue) and mean temperature (green) for Feira de Santana (FSA). Climate data available as Dataset 1. (B) Resulting Bayesian MCMC fit to weekly (black line: data, purple line: model fit) and cumulative incidence (black line: data, grey line: model fit). (A,B) The grey areas highlight the period before the Zika outbreak, the white areas highlight the period for which notified case data was available, and the yellow shaded areas highlight the period for which mean climatic data was used (see Materials and Methods). (C) Climatic series as in A and estimated $R_0$ for the period of the outbreak (2015–2017) ($R_0$ absolute values in *Figure 2—figure supplement 3*). (D) Correlations between the estimated $R_0$ and climatic variables (intercepts: 0.839 for humidity, 0.067 for rainfall and 0.658 for temperature). (E) Correlations between the case counts and climatic variables (intercepts: 0.487 for humidity, 0.024 for rainfall and 0.862 for temperature). (D,E) Points presented are from timepoints (weeks) for which incidence was notified. (A–E) Y-axis normalised between 0 and 1 for visualisation purposes.

DOI: https://doi.org/10.7554/eLife.29820.006

The following figure supplements are available for figure 2:

**Figure supplement 1.** Relationship between temperature and egg hatching success.
DOI: https://doi.org/10.7554/eLife.29820.007

**Figure supplement 2.** Prior selection and sensitivity.
DOI: https://doi.org/10.7554/eLife.29820.008

**Figure supplement 3.** Eco-epidemiological factors and model fit to notified cases.
DOI: https://doi.org/10.7554/eLife.29820.009

humidity and mean temperature data in FSA between 2013 and May 2017 (*Figure 2A*). The data showed erratic fluctuations in rainfall with sporadic episodes of intense rain but without a clear seasonal trend. Temperature, on the other hand, presented a much clearer seasonal signature with fixed amplitudes between 22 and 27 degree Celsius, peaking between December and May. Humidity showed an intermediate scenario and appeared correlated with periods of intense rainfall but negatively correlated with temperature.

By fitting our climate-driven transmission model to the local climate and ZIKV case data (see Material and Methods and *Figure 2B*) we obtained parameter estimates for the mosquito lifespan as well as the viral extrinsic incubation period (EIP) for the same period. Mosquito lifespan and EIP are main drivers of vectorial capacity and both showed seasonal oscillations with median values of around 9 and 5 days, respectively (*Figure 2—figure supplement 3*), which are in line with ranges found in the literature (*Trpis et al., 1995*; *Trpis and Hausermann, 1986*; *Andraud et al., 2012*; *Hugo et al., 2014* and *Table 3*). Importantly, there was a strong negative temporal correlation between these two variables, with periods of longer EIP coinciding with shorter lifespans and vice-versa. This negative relationship resulted in large temporal variations in vectorial capacity and thus seasonal oscillations in the daily reproductive numbers, $R_0$, with a median value of 2.7 in the period 2015–2017 (range 1.0–4.3, *Figure 2—figure supplement 3*), and 2.2 before 2015, peaking in the local summer months between December and April (*Figure 2C*). Importantly, $R_0$ remained above 1 for the entire period, indicating a high suitability for ZIKV in FSA. It should be noted that $R_0$ in this context is a time-dependent variable, i.e. $R_0(t)$, but out of convenience we simply refer to it as $R_0$.

We also looked at the relationship between each climatic variable and $R_0$ and case counts (*Figure 2D and E*, respectively). The transmission potential was strongly and positively correlated with temperature ($r^2 = 0.728$) and negatively with humidity ($r^2 = 0.26$). As expected, from the highly random patterns in the climate series, there was no correlation between $R_0$ and rainfall ($r^2 = 0.008$). In contrast, there was an opposite trend in the relationship between the climatic variables and case counts, with a positive correlation with humidity ($r^2 = 0.28$) and a negative correlation with temperature ($r^2 = 0.23$). As with $R_0$ there was only a weak observable trend in the relationship between rainfall and the number of Zika cases. It should be understood that this macroscopic analysis does not take into account the expected temporal lags due to mosquito development, incubation periods etc., so the purpose here was simply to identify a general qualitative relationship between climate, vectorial capacity and disease incidence.

## Model fit and parameter estimates

Four parameters of public health importance were estimated by our MCMC framework: the date of introduction, the human infectious period, the human (intrinsic) incubation period, and the case observation rate (*Table 4*). The posterior for the introduction date showed a strong support for an introduction into FSA in early-mid December 2014 (estimated median: $10^{th}$ of December), i.e. around 7–8 weeks before the first notified case (*Figure 3A*). The estimated human infectious period was $\approx 6$ days (*Figure 3C*, median = 5.9, 95% CI [5.47–6.14]), which was very similar to the estimated incubation period (*Figure 3D*, median = 5.8, 95% CI [5.6–6.15]) and in line with previously estimated ranges for ZIKV (*Table 3*). In this context it is important to note that informative priors had been used for these 2 parameters (*Figure 2—figure supplement 2.*), and the posterior for the incubation period presented an adjustment of $\approx -0.5$ days relative to the proposed distribution from the literature.

**Table 3.** Literature-based reports on key ZIKV epidemiological and entomological parameters.

| Parameter/Function | Values and ranges reported | References |
|---|---|---|
| Intrinsic incubation period | 6.5, 5.9 days | [ 34, 50 ] |
| Human infectious period | 4.7, 9.9 days | [ 34, 50 ] |
| Extrinsic incubation period | 8.2, 10, 7 days | [ 34, 51, 84 ] |
| Attack rates | 74, 50, 73, 94, 52% | [ 17, 26, 47, 28 ] |
| $R_0$ | 3.2, 2.5, 4.8, 2.05, 2.6–4.8, 4.3–5.8, 1.8–2.0 | [ 17, 37, 38, 47, 64 ] |
| Observation rate | 0.024, 0.06, 0.03, 0.11 | [ 17, 37, 47 ] |

DOI: https://doi.org/10.7554/eLife.29820.010

**Table 4.** Model estimated parameters.

| Notation | Description | Ranges / priors |
|---|---|---|
| $t_0$ | time point of first case (in a human) | $(\infty, \infty)$ |
| $K$ | aquatic carrying capacity | $(0, \infty)$ |
| $\eta$ | linear factor for mosquito mortality | $(0, \infty)$ |
| $\alpha$ | linear factor for extrinsic incubation period | $(0, \infty)$ |
| $\rho$ | non-linear factor for effects of humidity and rainfall | $(0, \infty)$ |
| $\sigma^h$ | human infectious period | $(0, 15)$ |
| $\gamma^h$ | human (intrinsic) incubation period | $(0, 15)$ |
| $\zeta$ | observation rate | $(0, 1)$ |

DOI: https://doi.org/10.7554/eLife.29820.011

Of particular interest here was the very low observation rate (*Figure 3B*), with a median of just under 0.004 (median = 0.0039, 95% CI [0.0038–0.0041]), which equates to less than 4 in 1000 infections having been notified during the epidemic in FSA. Although lower than other previously reported estimates, this would explain the relatively long period of low viral circulation before the epidemic took off in April 2015. That is, based on our estimates, there were around 2,700 Zika infections during the first 2 months, of which only 10 were notified. More importantly, when applying this rate to the total number of cases we found that by the end of the first epidemic wave around 65% (95% CI [57.0–72.9]) of the population in FSA had been infected by the virus. This high attack rate is not unusual for Zika, however, and is in general agreement with observations elsewhere (*Table 3*).

## Future transmission potential for Zika virus

As illustrated by the cumulative attack rate in *Figure 4A*, and similar to estimates from other regions in the world (*Table 3*), nearly 65% of the population got infected by ZIKV by the end of 2015, which rose to over 75% (95% CI [76.9–84.3]) by the end of 2016. During the first wave most cases occurred off-season, here defined by our estimated daily reproductive number ($R_0$), while the second wave appeared much more synchronized with the period of high transmission potential. Notably, this temporal phenomenon has also been observed for the chikungunya virus (CHKV) when it was first introduced into FSA in 2014 (*Faria et al., 2016b*).

The amassed accumulation of herd-immunity during the first wave resulted in a marked difference between the estimated basic reproductive number, $R_0$, and the effective reproductive ratio ($R_e$) by the end of 2015 (*Figure 4A*). This in turn might explain the marked reduction in Zika cases in FSA in 2016, at a time when the virus was infecting large numbers of individuals elsewhere in the country (*Figure 1A,C*). At the start of 2016, $R_e$ was estimated to be more than 3 times smaller than $R_0$, which increased to 5 by the beginning of 2017. Projecting into the future using average climate data for this region showed that the mean effective reproductive number is expected to remain low and close to 1 for the next few years, suggesting a very weak potential for ZIKV endemicity in the near future. In fact, the sporadic nature of Zika cases in 2017 strongly suggest that herd immunity in this region is at a sufficiently high level to prevent sustained transmission. Furthermore, during 2017, $R_e$ was on average less than 1 (mean: 0.62, range: 0.25–1.06), and we would therefore argue that the small number of cases (1.4% of 2015–2017) were mostly a result of small transmission chains, either from resonant transmission from the previous year, or from introduction events from nearby locations. Crucially, this would also explain why our ODE model matched both the dynamics and the sizes of the first two epidemic waves in FSA between 2015 and 2016 but failed to capture the small number of cases during 2017 (*Figure 2B*).

Without external introductions of infectious individuals (human or vector) our results predicted an epidemic fade-out by 2017, in accordance with the lack of notified cases after March 2017 (*Figure 4A*). We therefore projected ZIKV's epidemic potential over the next two decades (until 2040) using stochastic simulations (see Material and Methods) while assuming different rates of viral introduction (*Figure 4B,C*). Our results showed that the potential for ZIKV to cause another outbreak or to establish itself endemically in FSA is strongly dependent on the frequency of re-introductions, whereby higher rates of external introductions might in fact help to sustain high levels of herd

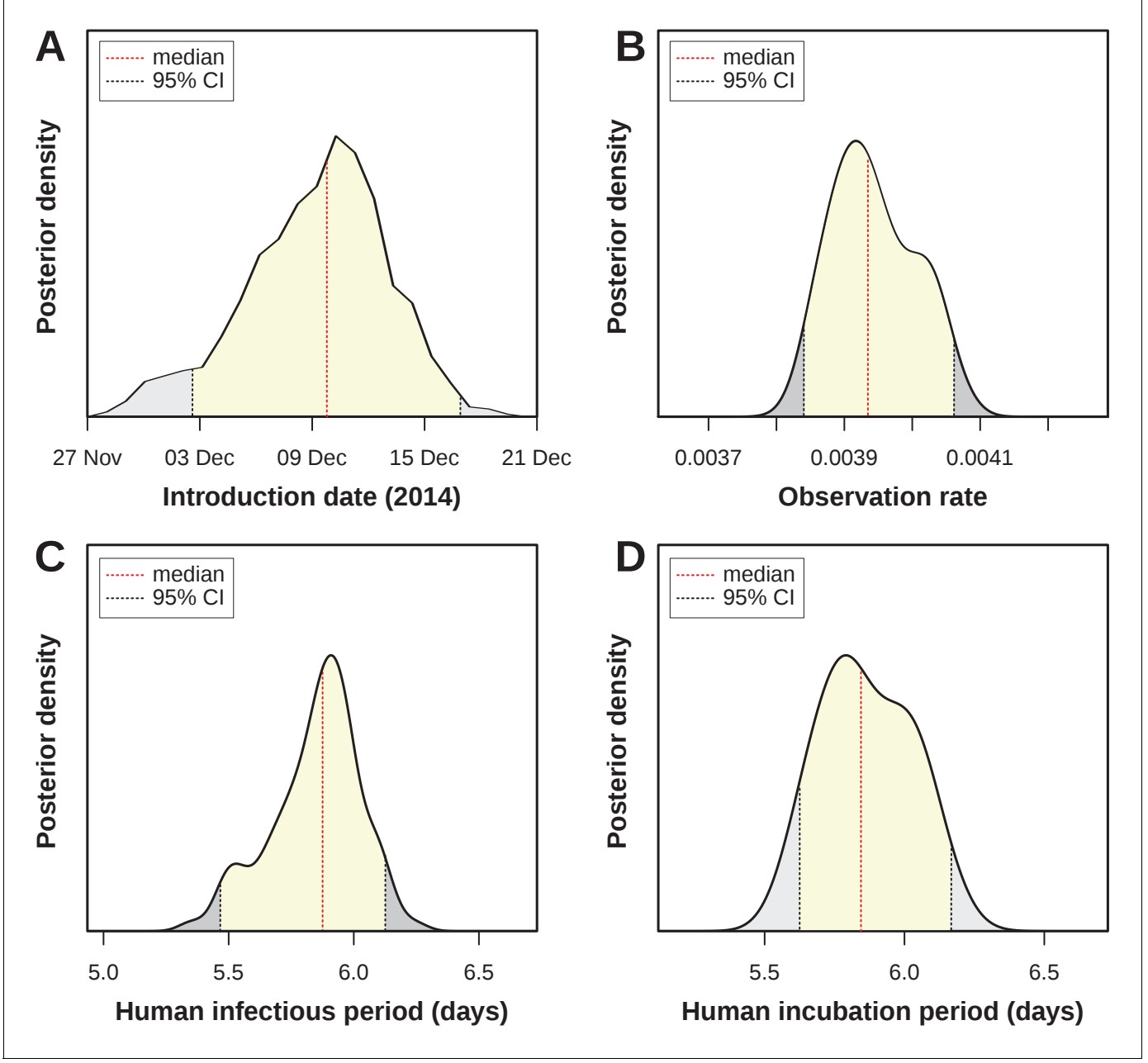

**Figure 3.** Estimated epidemiological and ecological parameters. MCMC posterior distributions, based on model fitting to notified case data between 2015–2017 and obtained from sampling 1 million MCMC steps after burn-in. (**A**) Posterior of the introduction date with median $10^{th}$ December 2014 (95% CI [01–16 Dec]). (**B**) Posterior of the observation rate with median 0.0039 (95% CI [0.0038–0.0041])). (**C**) Posterior of the human infectious period with median 5.9 days (95% CI [5.47–6.14]). (**D**) Posterior of the human (intrinsic) incubation period with median 5.8 days (95% CI [5.6–6.15]). Representative samples of 500 MCMC chain states are available in *Supplementary files 1–6*. See *Figure 3—figure supplement 1* for sample chain behaviour.

DOI: https://doi.org/10.7554/eLife.29820.012

The following figure supplements are available for figure 3:

**Figure supplement 1.** Sensitivity output for MCMC chains.
DOI: https://doi.org/10.7554/eLife.29820.013

**Figure supplement 2.** Eco-epidemiological factors and model fit to notified cases when using 2 observation rates.
DOI: https://doi.org/10.7554/eLife.29820.014

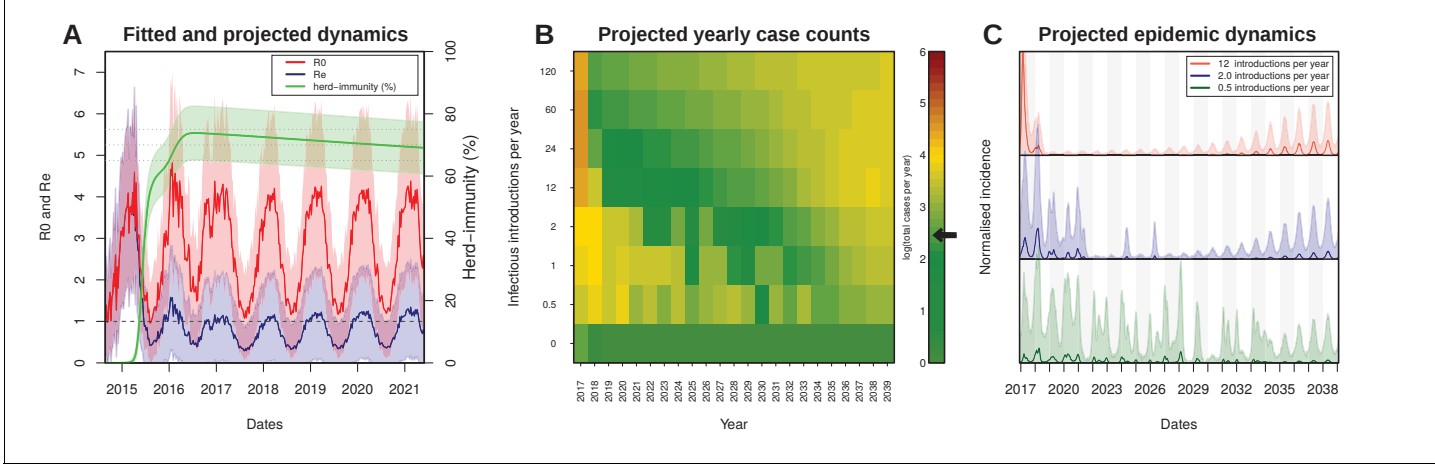

**Figure 4.** Projected Zika virus dynamics and transmission potential. (**A**) Fitted and projected epidemic attack rate (% population infected, or herd-immunity, green), basic reproduction number ($R_0$, red) and effective reproduction number ($R_e$, blue).(**B**) Colourmap showing the projected total number of annual cases depending on rate of external introduction of infectious individuals.The black arrow in the color scale marks the total number of real cases necessary for 1 notified case to be reported in FSA. (**C**) Projected incidence dynamics when considering less than 1 (green), 2 (blue) and 12 (red) external introductions per year. Grey and white shaded areas delineate different years. The Y-axes are normalised to 1 in each subplot for visualisation purposes. In (**B, C**) results are based on 1000 stochastic simulations with parameters sampled from the posterior distributions (*Figure 3*). Representative model solutions for incidence, R0 and Re from 500 MCMC chain samples are available in *Supplementary files 1–6* (both deterministic and stochastic).

DOI: https://doi.org/10.7554/eLife.29820.015

immunity, whereas infrequent introductions are more likely to result in notable outbreaks. That is, semi-endemic behaviour was only observed in simulations with low introduction rates (*Figure 4B–C*), as these scenarios strike a fine balance between a low number of new cases affecting herd-immunity levels and population turnover. In contrast, high introduction rates quickly exhaust the remaining susceptible pool, resulting in very long periods without epidemic behaviours.

## Sensitivity to reporting and microcephaly risk

In effect, our estimated observation rate entails the proportion of real infections that would have been notified if symptomatic and correctly diagnosed as Zika. Based on the previously reported Yap Island epidemic of 2007 (*Duffy et al., 2009*), the percentage of symptomatic infections can be assumed to be close to 18%. Unfortunately, measures of the proportion of individuals seeking medical attention and being correctly diagnosed do not exist for FSA, although it is well known that correct diagnosis for DENV is imperfect in Brazil (*Silva et al., 2016*). We therefore performed a sensitivity analysis by varying both the proportions of infected symptomatic individuals seeking medical attention and the proportion of those being correctly diagnosed for Zika. *Figure 5A* shows that if any of these proportions is less than 10%, or both between 15–20%, our observation rate of 3.9 per 1000 infections can easily be explained.

Finally we investigated the sensitivity of our results with regards to the expected number of newborns presenting microcephaly (MC). Following the observation that virtually all reported MC cases were issued before the summer of 2016 and with a lag of 5–6 months (*Figure 1A*), we assumed that the vast majority of Zika-associated MC cases would have been a consequence of the first epidemic wave in 2015. We used the estimated attack rate of approximately 65% from 2015 (*Figure 4A*) and varied the local birth rate and the theoretical risk of MC to obtain an expected number of cases. In agreement with other reports (*de Araújo et al., 2016*; *Cauchemez et al., 2016*; *Jaenisch et al., 2016*; *Johansson et al., 2016*), our model predicted a relatively low risk for MC given ZIKV infection during pregnancy (*Figure 5B,C*). In particular, using a conservative total of 21 confirmed MC cases in FSA, i.e. rejecting suspected or other complications, we estimate an average risk of approximately 0.35% of pregnancies experiencing ZIKV infection. Including the 3 foetal deaths where ZIKV infections were confirmed during pregnancy, i.e. using a total of 24 cases, only increased the risk to 0.39%. More generally, based on the results from our fitting approach and using the average birth

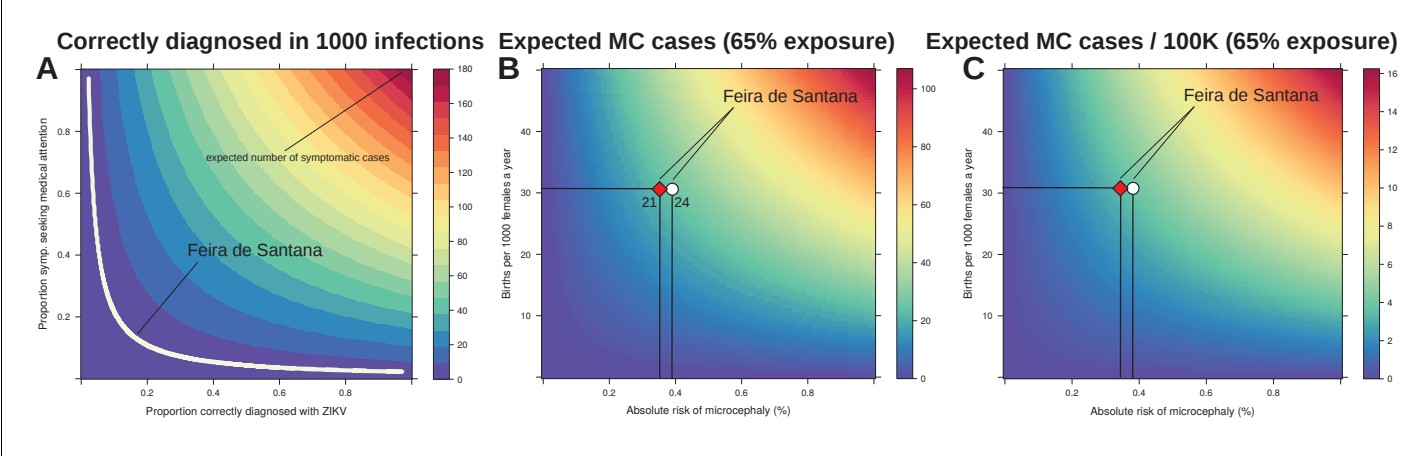

**Figure 5.** Sensitivity to reporting and microcephaly risk in Feira de Santana (FSA). (**A**) The observation rate (OR) can be expressed as the product of the proportion of cases that are symptomatic (0.18 [**Duffy et al., 2009**]), with the proportion of symptomatic that seek medical attention, and the proportion of symptomatic that upon medical attention get correctly diagnosed with Zika. In the white area the expected number of notified cases is the range obtained from fitting FSA case data (OR = 0.0039, 95% CI [0.0038–0.0041], **Figure 3**). (**B**) Expected number of cases of microcephaly (MC) for theoretical ranges of birth rate (per 1000 females) and risk of MC assuming 65% exposure of all pregnancies as estimated by our model for 2015 in FSA. (**C**) Expected number of MC per 100,000 individuals under the same conditions as in B. The symbols in B and C represent the total confirmed MC cases (21, red diamond), and the 21 MC plus 3 fetal deaths with confirmed Zika infection (24, white circle); the dashed horizontal line marks the number of births for FSA in 2015, and the vertical lines are the estimated risks per pregnancy.
DOI: https://doi.org/10.7554/eLife.29820.016

rates of FSA as guideline, we estimate that on average 3–4 MC cases are expected per 100 k individuals at 65% exposure to the virus.

## Discussion

Using an ento-epidemiological transmission model, driven by temporal climate data and fitted to notified case data, we analysed the 2015–2017 Zika outbreak in the city of Feira de Santana (FSA), in the Bahia state of Brazil and determined the conditions that led to the rapid spread of the virus as well as its future endemic and epidemic potential in this region. Given FSA's high suitability for ZIKV mosquito-vectors and its particular geographical setting as a state commerce and transport hub, our results should have major implications for other urban centres in Brazil and elsewhere.

The pattern of reported ZIKV infections in FSA was characterized by a large epidemic in 2015, in clear contrast to total reports at the country-level, peaking during 2016. Most notably for FSA was the epidemic decay in 2016 and fadeout in 2017. In order to resolve whether this was due to a lower transmission potential of ZIKV in 2016/2017 in FSA, we calculated the daily reproductive number ($R_0$) between 2013 and 2017 but found no notable decrease in 2016. Interestingly, the maximum $R_0$ in that period was observed in the season 2015/2016, coinciding with El Niño (**Golden Gate Weather Services, 2017**) and thus in line with the hypothesis that this phenomenon may temporary boost arboviral potential (**Caminade et al., 2017**; **van Panhuis et al., 2015**). By fitting our model to weekly case data we also estimated the observation rate, i.e. the fraction of cases that were notified as Zika out of the estimated total number of infections. It has previously been reported that the vast majority of Zika infections go unnoticed (**Table 3**), which is in agreement with our estimates of an observation rate below 1%. Based on this, around 65% of the local population were predicted to have been infected by ZIKV during the first wave in 2015, which is in the same range as the reported Zika outbreaks in French Polynesia (66%) (**Cauchemez et al., 2016**) and Yap Island (73%) (**Duffy et al., 2009**). The accumulation of herd-immunity caused a substantial drop in the virus's effective reproductive number ($R_e$) and hence a significantly lower number of cases during the second wave in 2016 and subsequent demise in 2017. In the context of FSA, it is possible that the high similarity of case definition to DENV, the concurrent CHIKV epidemic, and the low awareness of ZIKV at that time could have resulted in a significant number of ZIKV infections being classified as either dengue or

chikungunya. Furthermore, based on our analysis, we would argue that the percentage of correctly diagnosed ZIKV infections and infected individuals seeking medical attention must have been exceptionally low (both lower than 20%).

The age structure of notified cases showed a higher than expected incidence risk ratio (IRR) for individuals under the age of 4 years and a lower than expected risk for individuals aged + 50 years. This contrasts the observation during the Zika outbreak on Yap Island in 2007, where all age classes, except the elderly, presented similar attack rates (*Duffy et al., 2009*). We note here, however, that the Yap Island analysis was based on both a retrospective analysis of historical hospital records and prospective surveillance (serology, surveys). It is therefore possible that the signatures amongst the youngest and oldest individuals in FSA may reflect deficiencies and/or biases in local notified data. Such signatures could emerge by both a rush of parents seeking medical services driven by a hyped media coverage or prioritization of child-care due to the emergence of microcephaly during the Zika epidemic and a small proportion of the elderly seeking or having access to medical attention. In fact, the increased risk in young children in 2016 may have been a result of increased awareness as well as the interventions by the WHO in the second year. We also found a small increase in IRR in the 20–34 years age group, particularly during 2016, which could be indicative of the small contribution of sexual transmission (*Moghadas et al., 2017*; *Maxian et al., 2017*). Most of these observations are speculative, however, and more detailed data will be required to fully understand these age-related risk patterns. For instance, initiatives such as the ZiBRA Project (*ZIBRA, 2016*; *Faria et al., 2016c*; *Faria et al., 2017*), which perform mobile and real-time sampling with portable genome sequencing, could prove to be essential for a retrospective and future analysis of the ZIKV epidemic in Brazil, especially in areas where high levels of herd-immunity will prevent large-scale circulation in the coming years (*Ferguson et al., 2016*).

The implicit consideration of climate variables as drivers of vector biology allowed us to ascertain the relative roles of temperature, humidity and rainfall for Zika's basic and effective reproductive potentials ($R_0$ and $R_e$, respectively). Similar to other studies in temperate and tropical settings, we found that temperature, with its direct influence on mosquito lifespan, aquatic development and extrinsic incubation period, was the key driver of seasonal oscillations in the transmission potential (*Lourenço and Recker, 2014*; *Mordecai et al., 2016*; *Mourya et al., 2004*; *Feldstein et al., 2015*). Rainfall, on the other hand, only seemed to play a marginal role and we argue that it may be a relevant player for arboviral transmission mainly in tropical regions subject to intense rain seasons, such as areas in South East Asia (*Cuong et al., 2011*; *Hii et al., 2012*; *Xuan et al., 2014*). We also noted that the correlations between climatic variables and case counts were inverted when addressed against the transmission potential. For instance, while temperature was positively correlated with $R_0$ it was negatively correlated with Zika cases. This implies that the transmission potential is readily responsive to climatic variation but that the Zika epidemics in FSA showed a slight but expected delay in relation to the peak in transmission potential, with case numbers generally increasing after a stable period of maximum $R_0$, followed by epidemic peaks that tended to coincide with declining $R_0$. An interesting observation is that the 2015 epidemic peaked approximately 3 months after the estimated peak in the virus's transmission potential, whereas there was much higher synchrony during the second wave in 2016. The same behaviour has been described for the CHIKV outbreak in FSA in 2014–2015 and which has been linked to highly discordant spatial distributions between the first two epidemics (*Faria et al., 2016b*). It is likely that similar spatial effects (*Kraemer et al., 2017*) were present in FSA's ZIKV outbreaks. Unfortunately we did not have access to sufficiently detailed spatial data to explore this hypothesis further.

A phylogenetic analysis has proposed that the introduction of ZIKV into Bahia took place between March and September 2014, although without direct evidence for its circulation in FSA at that time (*Naccache et al., 2016*). Our estimated date of introduction showed support for a date in early-mid December 2014, a few months after the proposed introduction into Bahia and just over 7 weeks before the first case of Zika was notified in FSA. Similar periods between the first notification and estimated introduction often represent the time taken to complete one or more full transmission cycles (human-mosquito-human) before a cluster of cases is generated of sufficient size for detection by passive surveillance systems (*Lourenço and Recker, 2014*). The case data also shows a 2 months period after the first notification during which weekly case numbers remained extremely low. This long period was unexpected as persistent circulation of ZIKV could hardly be justified by the observed total of only 10 cases. Given our estimated observation rate, however, the number of ZIKV

infections during this time could have amounted to over 2700 actual cases. In April, the number of cases increased rapidly, coinciding with the Micareta festival, which we argue may have played a role in igniting the exponential phase of the epidemic by facilitating human-vector mixing as well as a more rapid geographical expansion.

After calibrating our model to the 2015–2017 epidemic, we projected the transmission of ZIKV beyond 2017 using stochastic simulations and average climatic variables for this region. Without the possibility of externally acquired infections, local extinction was very likely by 2017 due to the high levels of herd-immunity. According to our study, Zika's reproductive potential ($R_e$) reached its lowest point in 2017, and it is expected to remain low for the next couple of years, given the slow replenishment of susceptibles in the population through births. When explicitly modelling the importation of infectious cases our projections for the next two decades corroborated the conclusions of previous modelling studies that suggest a weak endemic potential for ZIKV after the initial exhaustion of the susceptible pool (*Ferguson et al., 2016*; *Kucharski et al., 2016*). However, our simulations also showed that the future epidemic behaviour is strongly dependent on the frequency of re-introductions, where sporadic and unpredictable epidemics could still be in the order of hundreds of cases. Furthermore, given our estimated observation rate for the 2015–2017 epidemic, passive surveillance systems are unlikely to fully detect the scale and occurrence of such small epidemics, missing their actual public health impact, and as such efforts should thus be placed to improve ZIKV detection and diagnosis in order to optimize the local reporting rates and potential for control.

Human sexual and vertical transmission of ZIKV is an important public health concern, especially within the context of potential Zika-associated microcephaly (MC) and other neurological complications in pre- and neonatals. With a total of over 10,000 live births in 2015 in FSA, our crude estimate for the risk of Zika-associated MC per pregnancy was below 4 cases per 100,000 individuals in a generalized population under an attack rate of 65%. As discussed elsewhere (*Cauchemez et al., 2016*), this risk is extremely low when compared to other known viral-associated complications, such as those caused by infections by cytomegalovirus (CMV) and the rubella virus (RV) (*De Santis et al., 2006*; *Naing et al., 2016*). It is therefore crucial to reiterate that what makes the ZIKV a public health concern is not necessarily the per pregnancy risk of neurological complications, but rather the combination of low risk with very high attack rates. Other studies have reported that the risk for complications during the $1^{st}$ trimester of gestation is higher than the one estimated here. For example, in the French Polynesia (FP) outbreak (*Cauchemez et al., 2016*), the risk associated with ZIKV infection during the $1^{st}$ trimester was 1%, while the overall, full pregnancy risk was 0.42%, similar to our FSA estimates. For the Yap Island epidemic, no microcephaly cases have been reported. With an estimated 24 births per 1000 females (census 2000 as in (*Duffy et al., 2009*)) and using an overall risk of approximately 0.4% per pregnancy, only 0–3 cases per 100,000 individuals would have been expected. However, the island's small population size (7391 individuals (*Duffy et al., 2009*)) together with a general baseline of 0–2 microcephaly cases per 100,000 in many areas of the world (*Johansson et al., 2016*; *Butler, 2016*; *EUROCAT, 2003*) would explain the absence of reported cases. It is also important to consider that a variety of birth defects have been found to be statistically associated with Zika virus infection during pregnancy, of which MC is one possible outcome. While the risk for birth defects per pregnancy is consistently reported to be high, estimations for the risk of MC vary considerably. For example, recent clinical trials (*Honein et al., 2017*; *Brasil et al., 2016*) suggested that the risk of Zika-associated MC could be an order of magnitude higher than the estimate reported in this or other previous studies (*Cauchemez et al., 2016*; *Duffy et al., 2009*). At this stage it is not possible to explain these differences, but it is tempting to speculate that other factors must influence either the actual or estimated risk. For example, there could be diagnostic biases or differences between epidemiological and clinical studies. Alternatively, viral or host genetic background, as well as the pre-exposition to other arboviruses may influence the absolute risk experienced by local populations or cohorts.

Official notification of Zika infections in Brazil started on the $1^{st}$ of January 2016, although cases were reported in many other regions in Brazil during 2015. It is therefore plausible that the observation rate changed upon official guidelines and that the capacity to accurately diagnose and report Zika infections could have been lower in 2015 compared subsequent years. To explore this, we reran our fitting approach allowing for a possible change in the observation rate for 2016 and onwards (*Figure 3—figure supplement 2*) and found a similar observation rate for 2015 (0.0039 versus 0.0034) as well as a similar attack rate between the two model variants. However, the estimated

observation rate for 2016 and beyond was $\approx 4$ times larger than for 2015, implying a positive change due to changes in the surveillance system. Nevertheless, only about 13–14 out of 1000 Zika cases were reported after the $1^{st}$ of January 2016. It is hard to discern where the positive changes took place, but we suggest the revised diagnosis guidelines may have increased the proportion correctly diagnosed while the proportion of symptomatic individuals visiting medical facilities did not change. It is also tempting to speculate that the 2015/2016 imbalance in reporting may have been a general phenomenon across Brazil. As described elsewhere, it is thus possible that FSA is a good example of states and urban centres that may have witnessed larger epidemics than reported in 2015 (*de Oliveira et al., 2017*). This, together with our conclusion that low MC risk with very high attack rates makes ZIKV a public health concern, could explain why most MC reports at the level of the country were in 2015 (*de Oliveira et al., 2017*), although for many regions the total reported number of ZIKV cases may have been surprisingly small that year.

There are certain limitations to our approach, many of which could be revisited when more detailed data becomes available. For example, we assumed homogeneous mixing between human and mosquito hosts but it is possible that spatio-temporal heterogeneities may have played a role in FSA. Furthermore, we have curated and integrated functional responses of key entomological parameters to temperature, rainfall and humidity variation, which were originally reported for dengue viruses. Our fitting approach is also dependent on notified case data and it is possible that the reported cases are not representative of the initial expansion of the virus, which may have thwarted the obtained posterior of the introduction date. Finally, our future projections for the endemic and epidemic potential of ZIKV are based on average climatic trends of past years and do not capture the occurrence of natural variation between years, in particular for years affected by major Southern American climate events, such as the El Niño (*Caminade et al., 2017*).

In this study we have addressed the local determinants of ZIKV epidemiology in the context of a major urban centre of Brazil. Our results imply that control and surveillance of ZIKV should be boosted and focused in periods of high temperature and during major social events. These factors could identify windows of opportunity for local interventions to mitigate ZIKV introduction and transmission and should be transferable to other areas for which both temperature data and community event schedules are available. We further confirm that the high transmission potential of ZIKV in urban centres can lead to the exhaustion of the local susceptible pool, which will in turn dictate the long-term epidemic and endemic behaviour of the virus. Depending on the rate of re-introduction, sporadic outbreaks are to be expected, although these will be unlikely to result in a notable increase in the number of microcephaly cases due to their limited sizes and low risk per pregnancy. Nonetheless, these local sporadic occurrences could still have important public health consequences, and we argue that much better diagnostics and reporting rates are required for local authorities to detect and respond to such events in the near future. Our integrated mathematical framework is capable of deriving key insights into the past and future determinants of ZIKV epidemiology and its findings should be applicable to other major urban centres of Brazil and elsewhere.

## Materials and methods

### Demographic and socio-economic setting

Feira de Santana (FSA) is a major urban centre of Bahia, located within the state's largest traffic junction, serving as way points to the South, the Southeast and central regions of the country. The city has a population of approximately 620.000 individuals (2015) and serves a greater geographical setting composed of 80 municipalities (*municipios*) summing up to a population of 2.5 million. Although major improvements in water supply have been accomplished in recent decades, with about 90% of the population having direct access to piped water, supply is unstable and is common practice to resort to household storage. Together with an ideal (tropical) local climate, these are favourable breeding conditions for species of the *Aedes* genus of mosquitoes, which are the main transmission vectors of ZIKV, CHIKV and the dengue virus (DENV) that are all co-circulating in the region (*Kraemer et al., 2015*; *Carlson et al., 2016*). FSA's population is generally young, with approximately 30% of individuals under the age of 20% and 60% under the age of 34. In the year of 2015, the female:male sex ratio in FSA was 0.53 and the number of registered births was 10352, leading to a birth rate standard measure of 31 new-borns per 1000 females in the population.

## Climate data

Local climatic data (rainfall, humidity, temperature) for the period between January 2013 and May 2017 was collected from the Brazilian open repository for education and research (BDMEP, Banco de Dados Meteorológicos para Ensino e Pesquisa) (*Brazil BDMEP, 1961*). The climate in FSA is defined as semi-arid (warm but dry), with sporadic periods of rain concetrated within the months of April and July. Between 2013 and 2015, mean yearly temperature was 24.6 celsius (range 22.5–26.6), total precipitation was 856 mm (range 571–1141), and mean humidity levels 79.5% (range 70.1–88.9%). Temperature, humidity and precipitation per day is available as Dataset 1.

## Zika virus notified case data

ZIKV surveillance in Brazil is conducted through the national notifiable diseases information system (Sistema de Informação de Agravos de Notificação, SINAN), which relies on passive case detection. Suspected cases are notified given the presence of pruritic maculopapular rash (flat, red area on the skin that is covered with small bumps) together with two or more symptoms among: low fever, or polyarthralgia (joint pain), or periarticular edema (joint swelling), or conjunctival hyperemia (eye blood vessel dilation) without secretion and pruritus (itching) (*Brazil SINAN, 2016*; *Brazil, 2016*). The main differences to case definition of DENV and CHIKV are the particular type of pruritic maculopapular rash and low fever (as applied during the Yap Island ZIKV epidemic (*Duffy et al., 2009*)). The data presented in *Figure 1* for both Brazil and FSA represents notified suspected cases and is available as Dataset 3 (please refer to the Acknowledgement section for sources). Here, we use the terms *epidemic wave* and outbreak interchangeably (but see (*Perkins et al., 2016*)).

## Microcephaly and severe neurological complications case data

A total of 53 suspected cases with microcephaly (MC) or other neurological complications were reported in FSA between January 2015 and February 2017. Using guidelines for microcephaly diagnosis provided in March 2016 by the WHO (as in (*Faria et al., 2016c*)), a total of 21 cases were confirmed after birth and follow-up. A total of 3 fetal deaths were reported for mothers with confirmed ZIKV infection during gestation but for which no microcephaly assessment was available. The first confirmed microcephaly case was reported on the 24[th] of November 2015 and virtually all subsequent cases were notified before August 2016 (with the exception of 2). The microcephaly case series can be found in Dataset 4.

## Ento-epidemiological dynamic model

The ordinary differential equations (ODE) model and the Markov-chain Monte Carlo (MCMC) fitting approach herein used are based on the framework previously proposed to study the introduction of dengue into the Island of Madeira in 2012 (*Lourenço and Recker, 2014*). We have changed this framework to relax major modelling assumptions on the mosquito sex ratio and success of egg hatching, have included humidity and rainfall as critical climate variables, and have also transformed the original least squares based MCMC into a Bayesian MCMC. The resulting framework is described in the following sections, in which extra figures are added for completeness.

The dynamics of infection within the human population are defined in *Equations 1-5*. In summary, the human population is assumed to have constant size ($N$) with mean life-expectancy of $\mu^h$ years, and to be fully susceptible before introduction of the virus. Upon challenge with infectious mosquito bites ($\lambda^{v \to h}$), individuals enter the incubation phase ($E^h$) with mean duration of $1/\gamma^h$ days, later becoming infectious ($I^h$) for $1/\sigma^h$ days and finally recovering ($R^h$) with life-long immunity.

$$\frac{dS^h}{dt} = \mu^h N - \lambda^{v \to h} - \mu^h S^h \tag{1}$$

$$\frac{dE^h}{dt} = \lambda^{v \to h} - \gamma^h E^h - \mu^h E^h \tag{2}$$

$$\frac{dI^h}{dt} = \gamma^h E^h - \sigma^h I^h - \mu^h I^h \tag{3}$$

$$\frac{dR^h}{dt} = \sigma^h I^h - \mu^h R^h \tag{4}$$

$$N = S^h + E^h + I^h + R^h \tag{5}$$

For the dynamics of the mosquito population (*Equations 6-10*), individuals are divided into two pertinent life-stages: aquatic (eggs, larvae and pupae, $A$) and adult females ($V$) as in (*Yang et al., 2009*). The adults are further divided into the epidemiologically relevant stages for arboviral transmission: susceptible ($S^v$), incubating ($E^v$) for $1/\dot{\gamma}^v$ days and infectious ($I^v$) for life. The ˙ (dot) notation is here adopted to distinguish climate-dependent entomological factors (further details in the following sections).

$$\frac{dA}{dt} = \dot{c}^v f \dot{\theta}^v \left(1 - \frac{A}{K(R+1)}\right) V - (\dot{\epsilon}_A^v + \dot{\mu}_A^v) A \tag{6}$$

$$\frac{dS^v}{dt} = \dot{\epsilon}_A^v A - \lambda^{h \to v} - \dot{\mu}_V^v S^v \tag{7}$$

$$\frac{dE^v}{dt} = \lambda^{h \to v} - \dot{\gamma}^v E^v - \dot{\mu}_V^v E^v \tag{8}$$

$$\frac{dI^v}{dt} = \dot{\gamma}^v E^v - \dot{\mu}_V^v E^v \tag{9}$$

$$V = S^v + E^v + I^v \tag{10}$$

Here, the coefficient $\dot{c}^v$ is the fraction of eggs hatching to larvae and $f$ the resulting female proportion. For simplicity and lack of quantifications for local mosquito populations, it is assumed that the sex ratio remains at 1:1 (i.e. $f = 0.5$). Moreover, $\dot{\epsilon}_A^v$ denotes the rate of transition from aquatic to adult stages, $\dot{\mu}_A^v$ the aquatic mortality, $\dot{\mu}_V^v$ the adult mortality, and $\dot{\theta}^v$ is the success rate of oviposition. The logistic term $(1 - \frac{A}{K(R+1)})$ can be understood as the ecological capacity to receive aquatic individuals (*Tran et al., 2013*), scaled by a carrying capacity term $K(R+1)$ in which K determines the maximum capacity and R is the local rainfall contribution (further details on following sections).

From *Equations 6-10*, the mean number of viable female offspring produced by one female adult during its life-time, i.e. the basic offspring number $Q$, was derived (*Equation 11*). Most parameters defining $Q$ are climate-dependent, and for fixed mean values of the climate variables (ex. mean rainfall $\bar{R}$), expressions were derived for the expected population sizes of each mosquito life-stage modelled ($A_0, V_0$) which are used to initialize the vector population (*Equations 12-13*).

$$Q = \frac{\dot{\epsilon}_A^v}{\dot{\epsilon}_A^v + \dot{\mu}_A^v} \frac{\dot{c} f \dot{\theta}^v}{\dot{\mu}_V^v} \tag{11}$$

$$A_0 = K(\bar{R}+1)\left(1 - \frac{1}{Q}\right) \tag{12}$$

$$V_0 = K(\bar{R}+1)\left(1 - \frac{1}{Q}\right)\frac{\dot{\epsilon}_A^v}{\dot{\mu}_V^v} \tag{13}$$

## Viral transmission

In respect to the *infected host-type* being considered, the vector-to-human ($\lambda^{v \to h}$) and human-to-vector ($\lambda^{h \to v}$) incidence rates are assumed to be, respectively, density-dependent and frequency-dependent (*Equations 14-15*). Here, $\dot{a}^v$ is the biting rate and $\dot{\phi}^{v \to h}$ and $\phi^{h \to v}$ are the vector-to-human and human-to-vector transmission probabilities per bite. Conceptually, this implies that (i) an increase in the density of infectious vectors should directly raise the risk of infection to a single human, while (ii) an increase in the frequency of infected humans raises the risk of infection to a mosquito biting at a fixed rate. The basic reproductive number ($R_0$) is defined similarly to previous modelling approaches (*Equation 16*) (*Wearing and Rohani, 2006*; *Lourenço and Recker, 2013*). We further derived an expression for the effective reproductive ratio ($R_e$, *Equation 17*), taking into account the susceptible proportion of the population in real-time.

$$\lambda^{v \to h} = \left(\dot{a}^v \dot{\phi}^{v \to h} I^v S^h / N\right) \propto I^v \tag{14}$$

$$\lambda^{h \to v} = \left(\dot{a}^v \dot{\phi}^{h \to v} I^h S^v / N\right) \propto I^h / N \tag{15}$$

$$R_0 = \frac{(V/N) \sim \dot{a}^v \sim \dot{a}^v \sim \dot{\phi}^{v \to h} \sim \phi^{h \to v} \sim \dot{\gamma}^v \sim \gamma^h}{\dot{\mu}_V^v (\sigma^h + \mu^h)(\gamma^h + \mu^h)(\dot{\gamma}^v + \dot{\mu}_V^v)} \tag{16}$$

$$R_e = (S^h / N) \times (S^v / N) \times R_0 / (V/N) \tag{17}$$

## Markov chain monte carlo fitting approach

For the fitting process, the MCMC algorithm by Lourenco et al. is here altered to a Bayesian approach by formalising a likelihood and parameter priors (*Lourenço and Recker, 2014*). For this, the proposal distributions (q) of each parameter were kept as Gaussian (symmetric), effectively retaining a random walk Metropolis kernel. We define our acceptance probability $\alpha$ of a parameter set $\Theta$, given model ODE output $y$ as:

$$\alpha = min\{1, \frac{\pi(y|\Theta^\star)p(\Theta^\star)q(\Theta^o|\Theta^\star)}{\pi(y|\Theta^o)p(\Theta^o)q(\Theta^\star|\Theta^o)}\}$$

(18)

where $\Theta^\star$ and $\Theta^o$ are the proposed and current (accepted) parameter sets (respectively); $\pi(y|\Theta^\star)$ and $\pi(y|\Theta^o)$ are the likelihoods of the ODE output representing the epidemic data given each parameter set; $p(\Theta^o)$ and $p(\Theta^\star)$ are the prior-related probabilities given each parameter set. We fit the Zika virus cumulative case counts per week, for which no age-related or geographical data is taken into consideration.

For computational reasons and based on a previous approach (*Dorigatti et al., 2013*), the likelihoods $\pi$ were calculated as the product of the conditional Poisson probabilities of each epidemic data ($d_i$) and ODE ($y_i$) data point:

$$\pi(y|\Theta) = \prod_{i=1}^{N}[Pr\{y_i = d_i\}]$$

(19)

Note, in this case where we have low cases numbers in a large population, the Poisson likelihood represents a reasonable approximation to the Binomial process, which is expected to underlie the observed data.

## Fitted parameters

With the MCMC approach described above, all combinations of the *open* parameters in the ODE system that most likely represent the outbreak are explored (*Table 4*). In summary, the MCMC estimates the distributions for: (1) the carrying capacity $K$, an indirect estimate of the number of adult mosquitoes per human; (2) time point of the first case $t_0$, assumed to be in a human; (3) a linear coefficient $\eta$ that scales the effect of temperature on aquatic and adult mortality rates; (4) a linear coefficient $\alpha$ that scales the effect of temperature on the extrinsic incubation period; (5) a non-linear coefficient $\rho$ that scales the effects of humidity and rainfall on entoi

mological parameters; (6) the human infectious period $1/\sigma^h$; and (7) the human incubation period $1/\gamma^h$.

By introducing the linear coefficients $\eta$ and $\alpha$, the relative effect of temperature variation on mortality and incubation is not changed per se, but instead the baselines are allowed to be different from the laboratory conditions used by Yang et al. (*Yang et al., 2009*). For solutions in which $\eta, \alpha \to 1$, the laboratory-based relationships are kept. For a discussion on possible biological factors that may justify $\eta$ and $\alpha$ please refer to the original description of the method in (*Lourenço and Recker, 2014*) and (*Brady et al., 2013*). Finally, the introduction of $\rho$ allows the MCMC to vary the strength by which entomological parameters react to deviations from local humidity and rainfall means. In practice, the effect of rainfall and humidity can be switched off when $\rho \to 0$ and made stronger when $\rho \to +\infty$ (details below).

Initial analysis of the MCMC output raised an identifiability issue between the human infectious period ($1/\sigma^h$) and the linear coefficient ($\eta$) that scales the effect of temperature on vector mortality ($\eta$ scales the baseline mortality without changes to the response of mortality to temperature). Hence, changes in both $\eta$ and $1/\sigma^h$ result in similar scaling effects on the transmission potential $R_0$ (*Equation 16*) and thus unstable MCMC chains for $\eta$ and $1/\sigma^h$, with the resulting posteriors appearing to be bimodal (for which there was no biological support). We addressed this issue by using informative priors for four parameters for which biological support exists in the literature: $\eta$, $1/\sigma^h$, $1/\gamma^h$, and $\alpha$. Gaussian priors were used with means and standard deviations taken from the literature (see *Figure 2—figure supplement 2*).

## Constant parameters

The framework described above has only 4 fixed parameters that are neither climate-dependent nor estimated in the MCMC approach (*Table 2*). Amongst these, $\phi^{h \to v}$ is the per bite probability of transmission from human-to-mosquito, which we assume to be 0.5 (*Lounibos and Escher, 2008*; *Mohammed and Chadee, 2011*); the sex ratio of the adult mosquito population $f$ is assumed to be 1:1 (*Lounibos and Escher, 2008*; *Mohammed and Chadee, 2011*); the life-expectancy of the human population is assumed to be an average of 75 years (*WHO, 2016c*); and the biting rate is taken to be on average 0.25 although with the potential to vary dependent on humidity levels (details below) (*Trpis and Hausermann, 1986*; *Yasuno and Tonn, 1970*).

## Climate-Dependent parameters

For each of the temperature-dependent entomological parameters, polynomial expressions are found de novo or taken from previous studies fitting laboratory entomological data with temperature (T) values used in Celsius. For rainfall (R) and humidity (U), positive or negative relationships to entomological parameters are introduced using simple expressions, with values used after normalization to $[0, 1]$. We assume that some parameters are affected by a combination of temperature with either rainfal or humidity, but take their effects to be independent. A list of climate-dependent parameters and references is found in *Table 1*.

Polynomials of 4th degree for the mortality ($\mu_A^v, \mu_V^v$) and success ovipositon ($\theta^v$) rates are taken from the study by Yang and colleagues under temperature-controlled experiments on populations of *Aedes aegypti* (*Equations 19-21*) (*Yang et al., 2009*). For aquatic to adult ($\epsilon_A^v$) rate we use the $7^{th}$ degree polynomial of the same study (*Equation 20*). For the relationship between the extrinsic incubation period ($1/\gamma^v$) and temperature we apply the formulation by Focks et al. which assumes that replication is determined by a single rate-controlling enzyme (*Focks et al., 1995*; *Schoolfield et al., 1981*; *Otero et al., 2006*) (Equation 24). The probability of transmission per mosquito bite ($\phi^{v \to h}$) is here modelled (Equation 25) as estimated by Lambrechts and colleagues (*Lambrechts et al., 2011*). Finally, the relationship between temperature and the fraction of eggs that successfully hatch ($c^v$) is estimated de novo (Equation 26) by fitting a $3^{rd}$ degree polynomial to *Aedes aegypti* and *albopictus* empirical data described by Dickerson et al. (see *Figure 2—figure supplement 1*) (*Dickerson, 2007*; *Mohammed and Chadee, 2011*).

$$\epsilon_A^v(T) = 0.131 - 0.05723T + 0.01164T^2 - 0.001341T^3 + 0.00008723T^4 \tag{20}$$
$$- 0.000003017T^5 + 5.153 \times 10^{-8}T^6 - 3.42 \times 10^{-10}T^7$$

$$\mu_A^v(T) = 2.13 - 0.3797T + 0.02457T^2 - 0.0006778T^3 + 0.000006794T^4 \tag{21}$$

$$\mu_V^v(T) = 0.8692 - 0.1599T + 0.01116T^2 - 0.0003408T^3 + 0.000003809T^4 \tag{22}$$

$$\theta^v(T) = -5.4 + 1.8T - 0.2124T^2 + 0.01015T^3 - 0.0001515T^4 \tag{23}$$

$$\gamma^v(T) = \frac{0.003359\frac{Tk}{298} \times \exp\left(\frac{15000}{R}\left(\frac{1}{298} - \frac{1}{Tk}\right)\right)}{1 + \exp\left(\frac{6.203 \times 10^{21}}{R}\left(\frac{1}{-2.176 \times 10^{30}} - \frac{1}{Tk}\right)\right)} \tag{24}$$

$$\phi^{v \to h}(T) = 0.001044T \times (T - 12.286) \times (32.461 - T)^{1/2} \tag{25}$$

$$c^v(T) = (-184.8 + 27.94T - 0.9254T^2 + 0.009226T^3)/100.0 \tag{26}$$

We normalise the time series of rainfall (R) and humidity (U), further using the mean normalised values ($\bar{R}, \bar{U}$) as reference for extreme deviations from the expected local tendencies (*Bicout and Sabatier, 2004*; *Tran et al., 2013*). Rainfall is assumed to affect positively the fraction of eggs that successfully hatch ($c^v$) (*Alto and Juliano, 2001*; *Rossi et al., 2015*; *Tran et al., 2013*; *Madeira et al., 2002*). A similar positive relationship is taken for the vector biting rate ($a^v$) and humidity levels (*Yasuno and Tonn, 1970*), in contrast to a negative effect on the adult mosquito mortality rate ($\mu_V^v$) (*Alto and Juliano, 2001*).

$$c^v(R) = (R - \bar{R})/\sqrt{1 + (R - \bar{R})^2} \tag{27}$$

$$a^v(U) = (U - \bar{U})/\sqrt{1 + (U - \bar{U})^2} \tag{28}$$

$$\mu_V^v(U) = \bar{U} - (U - \bar{U})/\sqrt{1 + (U - \bar{U})^2} \tag{29}$$

Below is the complete formulation for each entomological parameter in time (t), depending on the climatic variables for which relationships are assumed to exist, including the MCMC fitted linear ($\alpha, \eta$) and non-linear ($\rho$) factors described above.

$$\epsilon_A^v(t) = \epsilon_A^v(T) \tag{30}$$
$$\mu_A^v(t) = \eta \mu_A^v(T) \tag{31}$$
$$\mu_V^v(t) = \eta \mu_V^v(T)[1 + \mu_V^v(U)]^\rho \tag{32}$$
$$\theta^v(t) = \theta^v(T) \tag{33}$$
$$\gamma^v(t) = \alpha \gamma^v(T) \tag{34}$$
$$\phi^{v \to h}(t) = \phi^{v \to h}(T) \tag{35}$$
$$c^v(t) = c^v(T)[1 + c^v(R)]^\rho \tag{36}$$
$$a^v(t) = a^v[1 + a^v(U)]^\rho \tag{37}$$

## Stochastic formulation of the ento-epidemiological model

A stochastic version of the ento-epidemiological framework was developed by introducing demographic stochasticity in the transitions of the dynamic system. This followed the original strategy described in (*Lourenço and Recker, 2014*), in which multinomial distributions are used to sample the effective number of individuals transitioning between classes per time step. Multinomial distributions are generalized binomials - $Binomial(n, p)$ - where $n$ equals the number of individuals in each class and $p$ the probability of the transition event (equal to the deterministic transition rate). This approach has also been demonstrated elsewhere (*Lampoudi et al., 2009*).

## Source code

The approach used in this study uses code in C/C++, bash and R scripts and is available at https://github.com/lourencoj/ArboWeD2/tree/ArboWeD2_V1. A copy is archived at https://github.com/elifesciences-publications/ArboWeD2.

## Acknowledgements

We are most grateful to Wanderson Klebeler de Oliveira, Livia Carla Vinhal, Mariana Pastorello Verotti, Giovanini Evelim Coelho and Claudio Maierovitch Pessanha Henrique from the Brazilian Ministry of Health for providing epidemiological data regarding Zika virus notified cases in Brazil. MML and EMC curated the Zika virus notified cases in Feira de Santana. JL and ASW received funding from the European Research Council under the European Union's Seventh Framework Programme (FP7/2007-2013)/ERC grant agreement no. 268904 – DIVERSITY. MR was supported by a Royal Society University Research Fellowship. The European Research Council under the European Union's Seventh Framework Program (FP7/2007-2013)/ERC grant agreement no. 614725-PATHPHYLODYN funded OP. MUGK's contribution was made possible by the generous support of the American people through the United States Agency for International Development Emerging Pandemic Threats Program-2 PREDICT-2 (Cooperative Agreement No. AID-OAA-A-14–00102). CJVA was supported by a fellowship from the Labex EpiGenMed, via the National Research Agency, Program for Future Investment and University of Montpellier [ANR-10-LA-12–01]. BL received funding from the Engineering and Physical Sciences Research Council (EPSRC) in the UK. NRF was supported by a Sir Henry Dale Fellowship jointly funded by the Wellcome Trust and the Royal Society (grant number 204311/Z/16/Z). The funders had no role in study design, data collection and analysis, decision to publish, or preparation of the manuscript.

## Additional information

### Funding

| Funder | Grant reference number | Author |
| --- | --- | --- |
| European Research Council | 268904 - DIVERSITY | José Lourenço Andrew Walker |

| Wellcome Trust and Royal Society | 204311/Z/16/Z | Nuno Rodrigues Faria |
| --- | --- | --- |
| International Development Emerging Pandemic Threats Program-2 | AID-OAA-A-14-00102 | Moritz UG Kraemer |
| Labex EpiGenMed | ANR-10-LABX-12-01 | Christian Julian Villabona-Arenas |
| Engineering and Physical Sciences Research Council | | Ben Lambert |
| European Research Council | 614725-PATHPHYLODYN | Oliver G Pybus |
| Royal Society | | Mario Recker |

The funders had no role in study design, data collection and interpretation, or the decision to submit the work for publication.

## Author contributions

José Lourenço, Conceptualization, Data curation, Formal analysis, Methodology; Maricelia Maia de Lima, Mario Recker, Conceptualization, Data curation, Investigation, Writing—original draft, Writing—review and editing; Nuno Rodrigues Faria, Conceptualization, Resources, Supervision, Investigation, Writing—original draft, Writing—review and editing; Andrew Walker, Data curation, Funding acquisition, Project administration; Moritz UG Kraemer, Erenilde Marques de Cerqueira, Conceptualization, Data curation; Christian Julian Villabona-Arenas, Ben Lambert, Conceptualization, Investigation, Methodology; Oliver G Pybus, Resources, Data curation, Supervision; Luiz CJ Alcantara, Conceptualization, Data curation, Funding acquisition, Investigation, Writing—original draft, Project administration, Writing—review and editing

## Author ORCIDs

José Lourenço http://orcid.org/0000-0002-9318-2581
Nuno Rodrigues Faria http://orcid.org/0000-0001-8839-2798
Moritz UG Kraemer http://orcid.org/0000-0001-8838-7147
Christian Julian Villabona-Arenas http://orcid.org/0000-0001-9928-3968
Mario Recker http://orcid.org/0000-0001-9489-1315

## Decision letter and Author response

Decision letter https://doi.org/10.7554/eLife.29820.028
Author response https://doi.org/10.7554/eLife.29820.029

# Additional files

### Supplementary files

• Source data 1. Dataset 1. Climate time series for Feira de Santana (2013–2017), including daily temperature, humidity and rainfall (FSA_climate_series.xls).
DOI: https://doi.org/10.7554/eLife.29820.017

• Source data 2. Dataset 2. Age-related data for Feira de Santana (2015), including case counts per age, total population numbers and gender ratios (FSA_age_data.xls).
DOI: https://doi.org/10.7554/eLife.29820.018

• Source data 3. Dataset 3. Total weekly notified cases in Feira de Santana (2015–2017) (FSA_incidence_series.xls).
DOI: https://doi.org/10.7554/eLife.29820.019

• Source data 4. Dataset 4. Total notified cases of Microcephaly in Feira de Santana (2015–2017), including confirmed and suspected/not confirmed (FSA_Microcephaly_series.xls).
DOI: https://doi.org/10.7554/eLife.29820.020

• Supplementary file 1. Sample model deterministic solutions for incidence (Incidence_detsolution.csv).
DOI: https://doi.org/10.7554/eLife.29820.021

• Supplementary file 2. Sample model deterministic solutions for R0 (R0_detsolution.csv).
DOI: https://doi.org/10.7554/eLife.29820.022

• Supplementary file 3. Sample model deterministic solutions for Re (Re_detsolution.csv).
DOI: https://doi.org/10.7554/eLife.29820.023

• Supplementary file 4. Sample model stochastic solutions for incidence (Incidence_stosolution.csv).
DOI: https://doi.org/10.7554/eLife.29820.024

• Supplementary file 5. Sample model stochastic solutions for R0 (R0_stosolution.csv).
DOI: https://doi.org/10.7554/eLife.29820.025

• Supplementary file 6. Sample model stochastic solutions for Re (Re_stosolution.csv).
DOI: https://doi.org/10.7554/eLife.29820.026

• Transparent reporting form
DOI: https://doi.org/10.7554/eLife.29820.027

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
