## [Decision Letter]

[Editors’ note: a previous version of this study was rejected after peer review, but the authors submitted for reconsideration. The first decision letter after peer review is shown below.]

Thank you for submitting your work entitled "Epidemiological and ecological determinants of Zika virus transmission in an urban setting" for consideration by *eLife*. Your article has been favorably evaluated by a Senior Editor and three reviewers, one of whom, Mark Jit, is a member of our Board of Reviewing Editors.

Our decision has been reached after consultation between the reviewers. Based on these discussions and the individual reviews below, we regret to inform you that your work in its current form will not be considered further for publication in *eLife*.

The three reviewers agreed that this is an interesting paper that has the potential to set a standard for ecological-epidemiological analysis of Zika outbreaks in these settings. However, we had serious misgivings about the model fitting process, interpretation of input data and lack of detail about some outputs. Partly as a result of this, we were not persuaded by the findings either.

If these methodological issues could be addressed, and the results can be used to substantiate a much more compelling and convincing story about Zika in Brazil then we would be willing to consider a revised manuscript as a new submission, which may be sent to the same reviewers. However, at this stage we cannot guarantee that we will review a revised manuscript.

If the technical issues can be addressed but the main hypotheses cannot be sustained, then we still think the manuscript has merit and would encourage submitting it to a more specialised journal.

The specific technical issues that would need to be addressed are the following:

a) Issues around the model fit

- The bimodal posterior distribution for infectious and incubation period in Figure 3 suggest there could be an issue with the model fitting. Two possibilities come to mind: either a lack of parameter identifiability in the model itself, or poor mixing of the MCMC chains. We would suggest testing for each of these. What do the pairwise correlations between the posteriors look like? What is the effective sample size of the MCMC outputs for each parameter? According to Table 4, there are 8 parameters estimated – what do the posteriors look like for the other 4?

- In Figure 4, the paper states that a stochastic model was used, but this wasn't mentioned in the Materials and methods. How was stochasticity incorporated? In addition, what time of year were the infections introduced in Figure 3, and how was this value chosen? It seems to me timing would have a big effect on the number of cases, depending on whether introduction co-coincided with a high value of *R_e_*.

- In Figure 5, we did not understand why the entire region of 0-8 cases was shaded blue, rather than just a line representing 4/1000 infections (or perhaps a boundary region to represent the posterior distribution of the estimate).

- Figure 5 is hard to interpret. The colour gradient seems to have been selected so that the line appears to go through the central microcephaly data point of 27, which makes it difficult to identify which regions produce high and low case numbers. It is also not clear what Figure 5 adds, other than normalising the results by the population size – in which case, should the numbers not be 6.2 times smaller (as the population is 620,000)?

- Why was a Poisson likelihood used for the observation process (equation 19), rather than, say, a binomial distribution?

- Why was the polynomial simplified to a 3rd degree one in equation 20? What effect did this have on the model in practice? Similarly, why was a 3rd degree polynomial used to fit the data in Figure 2—figure supplement 1? What impact could this assumption have had on model results?

- Should the observation rate be time-independent? One would expect that surveillance (and health care seeking) would improve as awareness about Zika increased.

b) Issues around interpretation of data and results

- We would have liked to see more discussion around estimates for the α and rho parameters, which control the extent to which environmental factors influence entomological dynamics. What contribution did humidity and temperature have? What are the implications for analysis in other settings, e.g. with stronger or weaker seasonal effects?

- In the fifth paragraph of the Discussion, it seems a stretch to suggest that the model estimates could be consistent with an autumn 2014 introduction. The lower 95% credible interval in Figure 3 is given as 2nd Jan 2015. What proportion of the posterior density falls within the range of dates implied by phylogenetic data?

- The context of the results is seriously misleading about the epidemiology of ZIKV in the rest of Brazil, specifically comparing 2015 to 2016. The state of Bahia is unique in that substantial surveillance was done in 2015. Zika did not become nationally reportable until January 2016, when reported case numbers increased substantially. From both microcephaly reports and anecdotal information, however, it is clear that massive outbreaks occurred in other states in 2015 (e.g. Pernambuco). This is not a problem with the model per se, but it is a big limitation to the conclusion that FSA was different from the rest of Brazil, (e.g. Figure 1; Results, second paragraph and fourth paragraphs; Discussion, second paragraph). These sections should be rewritten to specifically address the uncertainty in national reporting (almost complete), remove national versus FSA comparison of the epidemiology, and highlight how this model and other models could tell us something about what likely happened in other places. We see that as one of the strengths of this approach and it is not explored at all.

- The finding about infants is notable, but also susceptible to a clear bias. It should be clearly noted that there may be an increased probability of reporting infants for whom care is often prioritized, both by families who would seek it and institutions who would provide it. This likely increased even more after recognition of the association with microcephaly. Furthermore, this dataset may offer a unique opportunity to assess changes in reporting as that became clear. This would be true for both infants and women of reproductive age.

- The information on microcephaly and GBS was insufficient. In the Introduction, it is stated that both were coincident with ZIKV incidence, but that seems unlikely given that both tend to lag behind incidence. These curves should by shown and discussed more specifically as this is a key component to understanding generalizability and the reliability of the data being used.

- There should be a bit more context of other work on ZIKV and climate; it's not accurate to say "the effects of local climate variables, such as temperature and rainfall, have not yet been explored in relation to Zika transmission." That's true in some ways, but not that generalizable. A number of studies are already cited, e.g. Bogoch et al., 2016; Zhang et al., 2016; Perkins et al., 2016; Messina et al., 2016. This manuscript should point out what is unique here.

- The observation rate estimate is very low. Lower than both Yap and French Polynesia. Is there other evidence that supports such a low rate? Limited surveillance? Is it alternatively possible that the epidemic was spatial heterogeneity actually resulted in a smaller epidemic that had a higher reporting rate, lower attack rate, yet nonetheless produced herd immunity effects?

- There should be more discussion about the risk of microcephaly. The comparisons to French Polynesia and Yap are great but there is a lot of other work that has been done, especially clinical studies: https://www.ncbi.nlm.nih.gov/pubmed/26943629 and https://www.ncbi.nlm.nih.gov/pubmed/27960197. It is especially important to understand why the estimates in this manuscript may be on the very low end of what is being reported elsewhere in studies specifically aimed at measuring that risk.

- The analysis suggests that most susceptibles become infected and then immune soon after the first wave of the epidemic in 2015. The second wave in 2016 has a much lower attack rate with a higher proportion of infants. However, there is potential for a new outbreak some years in the future (the exact time is difficult to determine because the x-axes in Figure 4 are incorrectly labelled I think). It would be useful to show the age distribution and predicted microcephaly incidence related to the later outbreaks. If these occur mainly in young children born after 2015 then the public health relevance may be minimal. This has wider implications – does it imply that the long-term public health impact of Zika is minimal once the virus has been established as an endemic childhood infection? These are obviously very large claims that are probably unsustainable from the model in its current state, but without further clarity about results they are obvious extrapolations that readers may make.

c) Issues around reproducibility

- As is normally the rule with *eLife* modelling papers, the model code, input data and results (including MCMC samples from the converged joint posterior distribution) needed to reproduce the figure should be included as supplementary data files. Public data from cited online sources may be moved, edited or removed in future, so it is important to include everything required to reproduce the descriptive and modelling analysis with the paper itself.

*Reviewer #1:*

This manuscript fits a model with entomological, epidemiological and climactic variables to data on Zika cases during the 2015/16 outbreaks in one city in Brazil. The model suggests that most susceptibles were infected during the 2015 wave which led to lower incidence in the 2016 wave, and would prevent further epidemics till some years in the future.

Some questions:

1) The analysis suggests that most susceptibles become infected and then immune soon after the first wave of the epidemic in 2015. The second wave in 2016 has a much lower attack rate with a higher proportion of infants. However, there is potential for a new outbreak some years in the future (the exact time is difficult to determine because the x-axes in Figure 4 are incorrectly labelled I think). It would be useful to show the age distribution and predicted microcephaly incidence related to the later outbreaks. If these occur mainly in young children born after 2015 then the public health relevance may be minimal. This has wider implications – does it imply that the long-term public health impact of Zika is minimal once the virus has been established as an endemic childhood infection?

2) However, it is not clear to me exactly how the model fit works, e.g. is the age dependent notification data even used or just the aggregated counts? It would be useful to give the actual likelihood function being used as equation (19) in the appendix is too general (e.g. we aren't told exactly what *y_i_* or *d_i_* are).

3) The relationship between transmission and climactic variables is established via a set of mechanistic equations linking variables governing vector life cycle with climate. While this is sophisticated, it would be useful to see a more conventional multi-variable regression approach, just to ensure that some obvious relationship has not been lost in the detail.

4) Should the observation rate be time-independent? One would expect that surveillance (and health care seeking) would improve as awareness about Zika increased.

5) In Figure 4, it is not clear whether the x-axis in panels A and C are in days or years.

*Reviewer #2:*

The authors present a transmission modelling analysis of Zika in Feira de Santana, Brazil. I think their broad approach is an important one – combining environmental data in a mechanistic model has the potential to reveal some valuable insights into Zika epidemiology. However, I had some concerns about the robustness of the model estimates, and interpretation of the results.

I have the following comments:

- The bimodal posterior distribution for infectious and incubation period in Figure 3 suggest there could be an issue with the model fitting. Two possibilities come to mind: either a lack of parameter identifiability in the model itself, or poor mixing of the MCMC chains. I would suggest testing for each of these. What do the pairwise correlations between the posteriors look like? What is the effective sample size of the MCMC outputs for each parameter? According to Table 4, there are 8 parameters estimated – what do the posteriors look like for the other 4?

- In Figure 4, the authors state they use a stochastic model, but this wasn't mentioned in the Materials and methods. How was stochasticity incorporated? In addition, what time of year were the infections introduced in Figure 3, and how was this value chosen? It seems to me timing would have a big effect on the number of cases, depending on whether introduction co-coincided with a high value of *R_e_*.

- In Figure 5 did not understand why the entire region of 0-8 cases was shaded blue, rather than just a line representing 4/1000 infections (or perhaps a boundary region to represent the posterior distribution of the estimate).

- I found Figure 5 hard to interpret. It seems the authors have selected a colour gradient so the line appears to go through the central microcephaly data point of 27, which makes it difficult to identify which regions produce high and low case numbers. It is also not clear to me what Figure 5 adds, other than normalising the results by the population size – in which case, should the numbers not be 6.2 times smaller (as the population is 620,000)?

- I would have liked to see more discussed of estimates for the α and rho parameters, which control the extent to which environmental factors influence entomological dynamics. What contribution did humidity and temperature have? What are the implications for analysis in other settings, e.g. with stronger or weaker seasonal effects?

- In the fifth paragraph of the Discussion, it seems a stretch to suggest that the model estimates could be consistent with an autumn 2014 introduction. The lower 95% credible interval in Figure 3 is given as 2nd Jan 2015. What proportion of the posterior density falls within the range of dates implied by phylogenetic data?

- In the fifth paragraph of the Discussion, the authors suggest they do not have access to spatial data, but Figure 1 indicates they do, at least at some level of resolution. Could they clarify why this is not suitable for exploring heterogeneities to support their discussion point?

- In the subsection “Viral Transmission”, what was the motivation for have density and frequency dependent transmission for vector-human and H-V transmission?

- Why was a Poisson likelihood used for the observation process (equation 19), rather than, say, a binomial distribution?

- Why was the polynomial simplified to a 3rd degree one in equation 20? What effect did this have on the model in practice? Similarly, why was a 3rd degree polynomial used to fit the data in Figure 2—figure supplement 1? What impact could this assumption have had on model results?

*Reviewer #3:*

The manuscript describes a detailed transmission model of the ZIKV epidemic in the city of Feira de Santa, Brazil. The work is well done and generally interesting, but there were several important shortcomings.

First, the context of the results is seriously misleading about the epidemiology of ZIKV in the rest of Brazil, specifically comparing 2015 to 2016. The state of Bahia is unique in that substantial surveillance was done in 2015. Zika did not become nationally reportable until January 2016, when reported case numbers increased substantially. From both microcephaly reports and anecdotal information, however, it is clear that massive outbreaks occurred in other states in 2015 (e.g. Pernambuco). This is not a problem with the model per se, but it is a big limitation to the conclusion that FSA was different from the rest of Brazil, (e.g. Figure 1; Results, second and fourth paragraphs; Discussion, second paragraph). These sections should be rewritten to specifically address the uncertainty in national reporting (almost complete), remove national versus FSA comparison of the epidemiology, and highlight how this model and other models could tell us something about what likely happened in other places. I see that as one of the strengths of this approach and it is not explored at all.

The finding about infants is notable, but also susceptible to a clear bias. It should be clearly noted that there may be an increased probability of reporting infants for whom care is often prioritized, both by families who would seek it and institutions who would provide it. This likely increased even more after recognition of the association with microcephaly. Furthermore, this dataset may offer a unique opportunity to assess changes in reporting as that became clear. This would be true for both infants and women of reproductive age.

I also felt the information on microcephaly and GBS was insufficient. In the Introduction, it is stated that both were coincident with ZIKV incidence, but that seems unlikely given that both tend to lag behind incidence. These curves should by shown and discussed more specifically as this is a key component to understanding generalizability and the reliability of the data being used.

There should be a bit more context of other work on ZIKV and climate; it's not accurate to say "the effects of local climate variables, such as temperature and rainfall, have not yet been explored in relation to Zika transmission." That's true in some ways, but not that generalizable. A number of studies are already cited, e.g. Bogoch et al., 2016; Zhang et al., 2016; Perkins et al., 2016; Messina et al., 2016. This manuscript should point out what is unique here.

The observation rate estimate is very low. Lower than both Yap and French Polynesia. Is there other evidence that supports such a low rate? Limited surveillance? Is it alternatively possible that the epidemic was spatial heterogeneity actually resulted in a smaller epidemic that had a higher reporting rate, lower attack rate, yet nonetheless produced herd immunity effects?

Lastly, there should be more discussion about the risk of microcephaly. The comparisons to French Polynesia and Yap are great but there is a lot of other work that has been done, especially clinical studies: https://www.ncbi.nlm.nih.gov/pubmed/26943629 and https://www.ncbi.nlm.nih.gov/pubmed/27960197. It is especially important to understand why the estimates in this manuscript may be on the very low end of what is being reported elsewhere in studies specifically aimed at measuring that risk.

[Editors’ note: what now follows is the decision letter after the authors submitted for further consideration.]

Thank you for resubmitting your work entitled "Epidemiological and ecological determinants of Zika virus transmission in an urban setting" for further consideration at *eLife*. Your revised article has been favorably evaluated by Prabhat Jha (Senior Editor) and three reviewers, one of whom is a member of our Board of Reviewing Editors.

The work and manuscript have greatly improved, and the reviewers are satisfied with the approach of the model. However, there are some remaining issues about the way aspects of the model and the results are described that need to be addressed before acceptance, as outlined below:

1) The relationship between the Zika epidemic in FSA vs. the rest of Brazil is still not very clear, although it is improved from before. For example, in the last paragraph of the Introduction it states that Zika peaked elsewhere in the country in 2016. That may be true for some locations, but data suggest otherwise for many of the most affected states. As the authors now note, Zika case surveillance changed substantially in 2016. Clearly higher case numbers are associated with this, but microcephaly numbers were much higher for many states as a result of the 2015 epidemics, so in fact many of the NE states in particular likely had much bigger epidemics in 2015, and FSA may be a very good representation of what happened in those states, not an anomaly as implied in the manuscript. The authors even found evidence of a 4-fold increase in reported FSA between 2015 and 2016. In my view, this is a very important finding and emphasizes how little we know about 2015, even in area with relatively strong surveillance.

While we agree that it is not very helpful to speculate about exactly when the Zika epidemic peaked in different parts of Brazil, we think the overall message is still misleading – it currently appears to suggest that the epidemic in FSA was earlier than the rest of the country when the quality of surveillance in 2015 is probably not good enough to make such conclusions. A more appropriate message could be that detailed analysis of FSA indicated a large epidemic in 2015, which could possibly have occurred in other places although this was not picked up by the limited surveillance at that time.

2) The addition of informative priors is sensible. It would also be helpful to have the posterior outputs so the main figures and analysis can be reproduced. This will enable groups working in other parts of Brazil (or indeed other countries) to make use of the analyses to compare to their findings. References should be given for the priors shown in Figure 2—figure supplement 2 – at the moment only the human incubation and infectious period are given priors.

It would also be helpful to have a brief descriptive file that states what data are used in which figure (if not the actual plotting code itself) – e.g. it is not clear where the age distribution in Figure 1 came from.

3) With regards to the actual code, we take the comment that this would run to many thousands of lines. We actually have facilities to host this on *eLife*, but we would be happy as an alternative if this was archived e.g. in a suitable GitHub repository.

4) The causal link between ZIKV infection and at least some congenital outcomes (e.g. microcephaly) is now clear from multiple studies in multiple locations. While much is left to be learned, we recommend being more direct in the second paragraph of the Introduction.

5) The term 'herd-immunity' seems a little imprecise as immunity has unlikely reached a level that is truly protective against invasion. Perhaps 'population-level immunity' would be more appropriate?

---

## [Author Response]

[Editors’ note: the author responses to the first round of peer review follow.]

[…] The specific technical issues that would need to be addressed are the following:a) Issues around the model fit- The bimodal posterior distribution for infectious and incubation period in Figure 3 suggest there could be an issue with the model fitting. Two possibilities come to mind: either a lack of parameter identifiability in the model itself, or poor mixing of the MCMC chains. We would suggest testing for each of these. What do the pairwise correlations between the posteriors look like? What is the effective sample size of the MCMC outputs for each parameter? According to Table 4, there are 8 parameters estimated – what do the posteriors look like for the other 4?

It is true that our ODE model is very complex, such that the fitting procedure is likely to suffer from a parameter identifiability issue, which may have led to the seemingly bi-modal posterior distribution. It is also true that there is a certain degree of covariance between the parameters, which again can cause non-normal posteriors. In the original manuscript we used uninformative priors for all of our parameters and relied on empirical observations as a validation of our dynamic model and fitting output – e.g. mean rates of latency and recovery, and R0 values close to reported ranges in the literature. In order to address the possible identifiability issue and shape of the posteriors we now used informed (Gaussian) priors for 4 out of 8 estimated parameters. This was based on relevant biological / epidemiological observations, including human infectious and latency periods and the eta and α factors that determine the vector life-span and incubation periods according to climatic variables.

The inclusion of informative priors resulted in better-behaved posteriors and removed the semi-bimodal shapes from the posteriors for the human infectious and incubation periods (Figure 3). It is important to note, however, that the updated outputs resulting from the addition of the informative priors, as well as the addition of a further 9 months of notified data, are consistent with the ones presented in the original manuscript (see other responses below for comparison of new results under the revised changes to the methodology as proposed by the Editors and Reviewers).

We have updated our Materials and methods section accordingly to accommodate for the inclusion of informative priors. Figure 2—figure supplement 2 and Figure 3—figure supplement 1 now illustrate the posteriors distributions and output from our MCMC chains, demonstrating their stability both in terms of sampling and the levels of correlation between each pair of parameters.

- In Figure 4, the paper states that a stochastic model was used, but this wasn't mentioned in the Materials and methods. How was stochasticity incorporated? In addition, what time of year were the infections introduced in Figure 3, and how was this value chosen? It seems to me timing would have a big effect on the number of cases, depending on whether introduction co-coincided with a high value of R_e_.

We apologize for not providing any description about the stochastic implementation, which we have now added to our revised manuscript. In summary, we used multinomial distributions to sample the effective number of individuals transitioning between classes per time step, as implemented in our previous work and elsewhere. A detailed description of this approach can now be found in the Materials and methods section.

With regards to the introduction of infected hosts, this is also modelled stochastically with a fixed migration rate (of infected individuals) per year, which is varied between the sensitivity scenarios presented in Figure 4. In effect, different simulations with the same migration rate will experience introduction events at different time points. The mean and variation presented in the stochastic model output (Figure 4) represent the outcome of thousands of simulations and therefore include all possible scenarios in which introductions may have occurred, covering periods of both high and low *R_e_*/ *R_0_*.

- In Figure 5, we did not understand why the entire region of 0-8 cases was shaded blue, rather than just a line representing 4/1000 infections (or perhaps a boundary region to represent the posterior distribution of the estimate).

We agree that the blue area in Figure 5 was not the clearest way to present our results. As suggested by the reviewer, we have now revised the figure to illustrate the exact area that represents Feira de Santana in the background of the sensitivity output (using the 95% CI of the estimated observation rate, taken from the posterior).

- Figure 5 is hard to interpret. The colour gradient seems to have been selected so that the line appears to go through the central microcephaly data point of 27, which makes it difficult to identify which regions produce high and low case numbers. It is also not clear what Figure 5 adds, other than normalising the results by the population size – in which case, should the numbers not be 6.2 times smaller (as the population is 620,000)?

The original Figure 5 was intended to illustrate the sensitivity between the total of MC cases reported and the model estimations, in which the colours were chosen to have the mean at the total of 27 cases (orange line) and variation around this value fading away. The plot would show that under a very strict MC case definition (21, red diamond) or including all suspected counts (36, blue square) the risk of MC per pregnancy is similarly low. This implies that the public health impact of Zika is not necessarily due to high risk for complications *per se* but rather the potentially explosive nature of Zika epidemics than can affect a significant number of pregnancies. The idea behind Figure 5 was just to complement Figure 5, showing normalised (per 100,000) MC case numbers. In this context, we confirm that the expected number was indeed 6.2 times smaller.

Following the comments from the reviewers, we have revised Figure 5. We now include a uniform colour scale in Figure 5 to help “identify which regions produce high and low case numbers”, and with the updated MC data set, which has been further curated to include solely confirmed and rejected MC reports, we have also updated the estimated risk per pregnancy based on confirmed cases only. With regards to Figure 5, we would still argue that it complements Figure 5, but we would be happy to leave it out of the manuscript if the reviewers / Editors feel it is superfluous.

- Why was a Poisson likelihood used for the observation process (equation 19), rather than, say, a binomial distribution?

The decision of using a Poisson likelihood was based on previous experience, computational cost and the fact that we fit a small number of cases per time step in a large population size, in which case the Poisson distribution is a reasonable approximation for the Binomial. Poisson-based likelihoods have also been used successfully in other modelling approaches, and we have now referenced and justified our decision accordingly in the main text.

- Why was the polynomial simplified to a 3rd degree one in equation 20? What effect did this have on the model in practice? Similarly, why was a 3rd degree polynomial used to fit the data in Figure 2—figure supplement 1? What impact could this assumption have had on model results?

The simplifications in equation 20 and 24 had been done purely for computational reasons but we appreciate that this could raise doubts about the sensitivity of our results on these approximations. To avoid such problems we have revised the equations, now using their original formulations, and have updated the Materials and methods section accordingly. We would like to note, however, that this did not have an effect on our results.

- Should the observation rate be time-independent? One would expect that surveillance (and health care seeking) would improve as awareness about Zika increased.

We agree that the observation rate is likely to have changed in time between 2015 and 2017. In fact, it has now become clear that the Brazilian authorities changed their surveillance system on the 1^st^ January 2016. We therefore explored the effect of using a semi time-dependent observation rate: one for 2015 (zeta) and a different one (zeta’) from 1^st^ January 2016 onwards.

We found that the inclusion of a second observation rate did not change the model fit and results for 2015 and only slightly for 2016. In particular, the posterior of the observation rate in 2015 had virtually the same distribution (mean zeta = 0.0039, mean zeta’ = 0.0034). This resulted in very similar attack rates for 2015 between the two model variants. Consequently, our general conclusions for the epidemic behaviour in 2016-2017 were the same, whereby high levels of herd-immunity generated by the epidemic outbreak in 2015 significantly reduced Zika’s effective reproductive number and hence incidence. In contrast, the posterior mean of the second observation rate (zeta’) showed a 4-fold increase, suggesting that the implementation of the new surveillance system helped to detect / correctly diagnose a higher number of Zika cases.

We have decided to keep the simpler model with one observation rate in the main results of the manuscript. We took this decision on the basis that the observation and attack rates for the year 2015 did not change and because these are the main drivers for the projected behaviour of Zika in the near future. We fully recognize the public health importance of increasing the observation rates due to changes in the surveillance system, but we believe that it should be mainly explored in the Discussion section and supported by Figure 3—figure supplement 2. We are naturally open for guidance and comments on this decision by the Editors and reviewers, however.

b) Issues around interpretation of data and results- We would have liked to see more discussion around estimates for the α and rho parameters, which control the extent to which environmental factors influence entomological dynamics. What contribution did humidity and temperature have? What are the implications for analysis in other settings, e.g. with stronger or weaker seasonal effects?

We agree that these topics are of great interest and currently lacking in the literature. We have significantly changed Figure 2 and now present two subplots on the relationship between the climatic variables and the observed case counts and transmission potential. We have further added a paragraph discussing these findings in our revised manuscript. However, we believe that a more in-depth exploration would require well-curated arboviral data (as the one presented here) from more than one location, potentially both urban and rural, and that this topic goes beyond the scope of our manuscript. (It is indeed something that we are planning to do as a separate research study in the future).

- In the fifth paragraph of the Discussion, it seems a stretch to suggest that the model estimates could be consistent with an autumn 2014 introduction. The lower 95% credible interval in Figure 3 is given as 2nd Jan 2015. What proportion of the posterior density falls within the range of dates implied by phylogenetic data?

We fully agree that the discussion about the introduction of Zika was highly speculative. In the context of the revised manuscript with changes in the updated dataset and addition of informative priors it is clear that an introduction dating back to early autumn 2014 is not supported. We have therefore removed this point from our revised manuscript.

- The context of the results is seriously misleading about the epidemiology of ZIKV in the rest of Brazil, specifically comparing 2015 to 2016. The state of Bahia is unique in that substantial surveillance was done in 2015. Zika did not become nationally reportable until January 2016, when reported case numbers increased substantially. From both microcephaly reports and anecdotal information, however, it is clear that massive outbreaks occurred in other states in 2015 (e.g. Pernambuco). This is not a problem with the model per se, but it is a big limitation to the conclusion that FSA was different from the rest of Brazil, (e.g. Figure 1; Results, second paragraph and fourth paragraphs; Discussion, second paragraph). These sections should be rewritten to specifically address the uncertainty in national reporting (almost complete), remove national versus FSA comparison of the epidemiology, and highlight how this model and other models could tell us something about what likely happened in other places. We see that as one of the strengths of this approach and it is not explored at all.

We had no intention to mark Feira de Santana (FSA) as a special case of the Brazilian Zika epidemic. The purpose of the graph in Figure 1 was simply to compare the epidemiological timeseries and to show that the epidemic in FSA peaked in 2015, in contrast to Brazil as a whole, where Zika peaked the following year; this does obviously not exclude other places where the epidemic could also have peaked in 2015. Furthermore, we believe that the behaviour described in this study is potentially ubiquitous to many major urban centres in Brazil and elsewhere. Hence, we do not agree that this is in any way misleading. In our revised version we have tried to clarify that this is not a comparison between FSA and other places in Brazil, which we agree would be very interesting to investigate in more detail, but which would require a different approach and more detailed data from different locations across the country.

- The finding about infants is notable, but also susceptible to a clear bias. It should be clearly noted that there may be an increased probability of reporting infants for whom care is often prioritized, both by families who would seek it and institutions who would provide it. This likely increased even more after recognition of the association with microcephaly. Furthermore, this dataset may offer a unique opportunity to assess changes in reporting as that became clear. This would be true for both infants and women of reproductive age.

We fully agree that the apparent increases or decreases in risks in some of the age groups can potentially be due to reporting bias. We tried to extract as much information out of the data as possible but unfortunately it was too limited to go into further details about the apparent risk differences. In our manuscript we explicitly state that “With the lack of more detailed data it was not possible to ascertain whether these findings indicated age-related risks of disease, age-dependent exposure risks or simply notification biases”. Furthermore, and akin to the reviewer’s comment, in the original Discussion we mentioned that “… such signatures could emerge by both a rush of parents seeking medical services driven by a hyped media coverage during the ZIKV epidemic, and a very small proportion of the elderly seeking or having access to medical attention”. The wording of this paragraph has now been revised to reflect better the reviewer’s comment.

- The information on microcephaly and GBS was insufficient. In the Introduction, it is stated that both were coincident with ZIKV incidence, but that seems unlikely given that both tend to lag behind incidence. These curves should by shown and discussed more specifically as this is a key component to understanding generalizability and the reliability of the data being used.

We agree that the text relating to microcephaly (MC) and Zika infections may have been misleading and potentially incomplete, due to not having had access to the time scales in the MC dataset. To clarify, we did not intend to state that MC and Zika infections should coincide in time, and in fact our work from March 2016 was one of the first to report a lag of several months between the two (Faria et al. (2016) Science 352:aaf5036)). We have now obtained MC counts in time up to May 2017 and have added this data to Figure 1 (unfortunately, as before, data on GBS was not available). By comparing the Zika and MC time series for Feira de Santana we can now detect a lag of about 5.5-6 months, which is slightly longer than our previous report. This, however, can be explained by the fact that the dates for MC cases are the actual dates of confirmation, which is usually done postpartum (30-60 days after delivery). Crucially, as shown in the revised Figure 1, we are now able to validate our original conclusion that most MC cases reported were a consequence of the high attack rate in 2015 and consequently the high number of pregnancies at risk during that year. We have changed the manuscript to include results and discussion on the new time scale of the MC epidemic in Feira de Santana.

- There should be a bit more context of other work on ZIKV and climate; it's not accurate to say "the effects of local climate variables, such as temperature and rainfall, have not yet been explored in relation to Zika transmission." That's true in some ways, but not that generalizable. A number of studies are already cited, e.g. Bogoch et al., 2016; Zhang et al., 2016; Perkins et al., 2016; Messina et al., 2016. This manuscript should point out what is unique here.

It is true that the studies we reference include some weather proxies or climate variables. However, in our manuscript we present an integrated, dynamic framework that explicitly models the life-cycle of the vector under climatic drivers, alongside viral transmission between mosquitoes and humans. Studies Bogoch et al., 2016, Perkins et al., 2016 and Messina et al., 2016 on the other hand, do not explore a dynamic transmission model and use instead climatic variables for geo-mapping and suitability indexation. The study Zhang et al., 2016 does include a dynamic model and climate-dependent parameters, and although the proposed framework offers a great opportunity to study the impact of climate on Zika transmission, this has not been explored. In order to not overstate the focus of our study we have changed the statement to: “Climate variables are critical for the epidemiological dynamics of Zika and other arboviral diseases, such as dengue [Gao et al., 2016; Bewick et al., 2016; Lourenço and Recker, 2014; Feldstein et al., 2015] and chikungunya [Kraemer et al., 2015; van Punhuis et al., 2011; Poletti et al., 2011]. Although these have also been previously addressed in mapping and / or modelling studies (e.g. [Bogoch et al., 2016; Zhang et al., 2016; Perkins et al., 2016; Messina et al., 2016]), their effects as ecological drivers for the emergence, transmission and endemic potential of the Zika virus, especially in the context of a well described outbreak, have not yet been addressed in detail.”

- The observation rate estimate is very low. Lower than both Yap and French Polynesia. Is there other evidence that supports such a low rate? Limited surveillance? Is it alternatively possible that the epidemic was spatial heterogeneity actually resulted in a smaller epidemic that had a higher reporting rate, lower attack rate, yet nonetheless produced herd immunity effects?

At this point we can only speculate but would argue that it is likely a combination of various factors. The first one is the already mentioned similarity in symptoms to other arboviral diseases, which, given the chikungunya outbreak that took place in 2014-2015, one season before the introduction of the Zika virus, could have led to a high number of misdiagnosed cases. Limited surveillance is certainly another big factor, and as shown in this study could have contributed to the large number of missed cases in 2015-2016. With regards to spatial effects, or rather the lack thereof in our ODE model approach, this is an interesting aspect and we agree that this could have led to an overall overestimation of the attack rate and hence underestimation of the observation rate (as is usually the case with ODE models). Unfortunately we do not have access to higher spatially resolved case data to explore this in more detail but have added this as a Discussion point in our revised version. Because of these uncertainties we present sensitivity analyses over the observation rate (Figure 5) as well as the number of microcephaly (MC) cases and the associated risk. In this context, we would like to note that model validation comes partially from estimated parameters. That is, although we leave certain parameters free in the model, these are obtained by the fitting procedure and crucially match expectations from the literature (e.g. human infectious period, mosquito life-span, etc.). Furthermore, we obtain a risk of MC per pregnancy that matches previous reports. We therefore argue that although it is possible that several factors are missing from our modelling approach, such small validations are reassuring in the sense that they match expectations and dictate that model calibration by the MCMC approach, based on reported data, fits biological and epidemiological expectations.

- There should be more discussion about the risk of microcephaly. The comparisons to French Polynesia and Yap are great but there is a lot of other work that has been done, especially clinical studies: https://www.ncbi.nlm.nih.gov/pubmed/26943629 and https://www.ncbi.nlm.nih.gov/pubmed/27960197. It is especially important to understand why the estimates in this manuscript may be on the very low end of what is being reported elsewhere in studies specifically aimed at measuring that risk.

We would like to note that our submission preceded the publication of these clinical trials, and thank the reviewer for their reference. In our study we estimate the risk of MC per pregnancy and find it to be consistent with observations in Yap Island and the French Polynesia. The clinical studies mentioned above do not focus on MC alone, but report instead a wide variety of birth defects and associated risks. The general impression from clinical studies is that the risk for birth defects upon Zika infection is indeed high. For instance, Honein et al. report that “Birth defects were reported in 9 of 85 (11%; 95% CI, 6%-19%) completed pregnancies with maternal symptoms or exposure”, while Brasil et al. found that “Among 117 live infants born to 116 ZIKV-positive women, 42% were found to have grossly abnormal clinical or brain imaging findings or both”. Notably, the particular risk of MC in both studies is much lower than the risk for general birth defects (4/117 for Brasil et al., and 18/442 for Honein et al.) but is still ~10 times larger than the risk reported in our manuscript and for the French Polynesia.

At this stage it is difficult, if not impossible, to resolve these differences, but it is tempting to speculate that other (known) factors must influence either the actual or the estimated risk. For example, there may be significant differences in the estimation of Zika attack rates in pregnant women versus the general population due to cross-reactivity of serological assays. There may also be diagnostic biases or differences between epidemiological and clinical studies. Viral or host genetic background, as well as the pre-exposition to other arboviruses, may also influence the absolute risk experienced by local populations, although it is hard to see how this could explain a ten-fold difference in risk. In the absence of a definite explanation we had to resolve to simply discuss this issue in the Discussion of our manuscript, but we would be interested to hear from the reviewer(s) if they had another, plausible explanation.

- The analysis suggests that most susceptibles become infected and then immune soon after the first wave of the epidemic in 2015. The second wave in 2016 has a much lower attack rate with a higher proportion of infants. However, there is potential for a new outbreak some years in the future (the exact time is difficult to determine because the x-axes in Figure 4 are incorrectly labelled I think). It would be useful to show the age distribution and predicted microcephaly incidence related to the later outbreaks. If these occur mainly in young children born after 2015 then the public health relevance may be minimal. This has wider implications – does it imply that the long-term public health impact of Zika is minimal once the virus has been established as an endemic childhood infection? These are obviously very large claims that are probably unsustainable from the model in its current state, but without further clarity about results they are obvious extrapolations that readers may make.

Our estimated cumulative attack rate of over 65% by the end of 2016 would suggest that major outbreaks are unlikely in the near future (the x-axes in Figure 4 have been updated). As our updated dataset shows, there have been a number of Zika cases in 2017, possibly through external introductions or through low-level background transmission events. The nature of these mini-outbreaks will have a big influence on herd-immunity levels in the population, which in turn will dictate, to a certain degree, when a new epidemic would become likely (as shown in Figure 4). However, the potential timing of a new outbreak is too speculative and depends on too many factors for any model to make accurate predictions.

If Zika was to become an endemic disease then the average age of infection would mostly be determined by its reproductive number, as with most other endemic diseases. One could even speculate that the number of MC cases would be relatively low, if the average age of infection was significantly below child bearing age, but that would depend on a much better knowledge about host immunity to Zika as well as the risk of MC in pre-exposed mothers (see discussion above). Given our current knowledge we would feel uncomfortable to make a strong statement about the future occurrence of MC. For these reasons we decided not to enter further into this discussion in our manuscript apart from mentioning that what makes the ZIKV a public health concern is not the per pregnancy risk of neurological complications, but rather the combination of low risk with very high attack rates. High numbers of MC cases should therefore mainly be a phenomenon of major epidemics observed in fully susceptible populations but unlikely in the context of endemic circulation.

c) Issues around reproducibility- As is normally the rule with eLife modelling papers, the model code, input data and results (including MCMC samples from the converged joint posterior distribution) needed to reproduce the figure should be included as supplementary data files. Public data from cited online sources may be moved, edited or removed in future, so it is important to include everything required to reproduce the descriptive and modelling analysis with the paper itself.

We are happy to provide the data and MCMC samples we analysed in this work as supplementary data files (done in the revised version), and we hope that our new Materials and methods section will be sufficient to allow the interested reader to reproduce the figures under the assumption they have the required computing skills. However, we would question the necessity, or even usefulness, to upload our entire computer code plus script files that contain thousands of lines of code and make use of various programming languages, and therefore like to ask the editor for clarification / guidance.

Reviewer #1:[…] Some questions:1) The analysis suggests that most susceptibles become infected and then immune soon after the first wave of the epidemic in 2015. The second wave in 2016 has a much lower attack rate with a higher proportion of infants. However, there is potential for a new outbreak some years in the future (the exact time is difficult to determine because the x-axes in Figure 4 are incorrectly labelled I think). It would be useful to show the age distribution and predicted microcephaly incidence related to the later outbreaks. If these occur mainly in young children born after 2015 then the public health relevance may be minimal. This has wider implications – does it imply that the long-term public health impact of Zika is minimal once the virus has been established as an endemic childhood infection?

Please see our response above.

2) However, it is not clear to me exactly how the model fit works, e.g. is the age dependent notification data even used or just the aggregated counts? It would be useful to give the actual likelihood function being used as equation (19) in the appendix is too general (e.g. we aren't told exactly what y_i_ or d_i_ are).

We apologize for not being clear on which data was used. The MCMC fits the ODE model output to the aggregated case counts and no age-related or spatial information is used. In fact, age-related information is not available in time and only as aggregated counts over an entire year (Figure 1). We have now made this clear in the Materials and methods section and explicitly mention that we fit case counts only, with no consideration of age or spatial information. We have further clarified that *y_i_* relates to an ODE data point and *d_i_* to an observed data point.

3) The relationship between transmission and climactic variables is established via a set of mechanistic equations linking variables governing vector life cycle with climate. While this is sophisticated, it would be useful to see a more conventional multi-variable regression approach, just to ensure that some obvious relationship has not been lost in the detail.

The relationship between climate variables and vector life cycle is taken from the published literature and based on experimental data, and we are unsure how a multi-variable regression approach, especially given the strong non-linear and non-monotonic relationships between many of the variables, could add to our methods / robustness of our results.

4) Should the observation rate be time-independent? One would expect that surveillance (and health care seeking) would improve as awareness about Zika increased.

Please see our response above.

5) In Figure 4, it is not clear whether the x-axis in panels A and C are in days or years.

We apologise for the poor labelling, this has been updated in a revised figure.

Reviewer #2:[…] I have the following comments:- The bimodal posterior distribution for infectious and incubation period in Figure 3 suggest there could be an issue with the model fitting. Two possibilities come to mind: either a lack of parameter identifiability in the model itself, or poor mixing of the MCMC chains. I would suggest testing for each of these. What do the pairwise correlations between the posteriors look like? What is the effective sample size of the MCMC outputs for each parameter? According to Table 4, there are 8 parameters estimated – what do the posteriors look like for the other 4?

Please see our response above.

- In Figure 4, the authors state they use a stochastic model, but this wasn't mentioned in the Materials and methods. How was stochasticity incorporated? In addition, what time of year were the infections introduced in Figure 3, and how was this value chosen? It seems to me timing would have a big effect on the number of cases, depending on whether introduction co-coincided with a high value of R_e_.

Please see our response above.

- In Figure 5 did not understand why the entire region of 0-8 cases was shaded blue, rather than just a line representing 4/1000 infections (or perhaps a boundary region to represent the posterior distribution of the estimate).

Please see our response above.

- I found Figure 5 hard to interpret. It seems the authors have selected a colour gradient so the line appears to go through the central microcephaly data point of 27, which makes it difficult to identify which regions produce high and low case numbers. It is also not clear to me what Figure 5 adds, other than normalising the results by the population size – in which case, should the numbers not be 6.2 times smaller (as the population is 620,000)?

Please see our response above.

- I would have liked to see more discussed of estimates for the α and rho parameters, which control the extent to which environmental factors influence entomological dynamics. What contribution did humidity and temperature have? What are the implications for analysis in other settings, e.g. with stronger or weaker seasonal effects?

Please see our response above.

- In the fifth paragraph of the Discussion, it seems a stretch to suggest that the model estimates could be consistent with an autumn 2014 introduction. The lower 95% credible interval in Figure 3 is given as 2nd Jan 2015. What proportion of the posterior density falls within the range of dates implied by phylogenetic data?

Please see our response above.

- In the fifth paragraph of the Discussion, the authors suggest they do not have access to spatial data, but Figure 1 indicates they do, at least at some level of resolution. Could they clarify why this is not suitable for exploring heterogeneities to support their discussion point?

The spatial data presented in Figure 1 is at the level of the *municipio*, equivalent to councils in the United Kingdom. The areas covered by municipios are large and extend far beyond the area of a city such Feira de Santana. Hence, while we agree that the data would be suitable for a *macro-spatial* analysis, it cannot be used to explore spatial heterogeneities at the level of an urban centre. Furthermore, the geographic data is aggregated by month, whereas in this study we are exploring the ecological and epidemiological determinant of Zika transmission at a much finer temporal scale.

- In the subsection “Viral Transmission”, what was the motivation for have density and frequency dependent transmission for vector-human and H-V transmission?

This is based on the following observations: for a vector-transmitted disease, an increase in the *density* of infectious vectors directly raises the risk of infection for a single human host. On the other hand, an increase in the *frequency* of infected humans raises the risk of infection to a single mosquito host, assuming fixed biting rates.

- Why was a Poisson likelihood used for the observation process (equation 19), rather than, say, a binomial distribution?

Please see our response above.

- Why was the polynomial simplified to a 3rd degree one in equation 20? What effect did this have on the model in practice? Similarly, why was a 3rd degree polynomial used to fit the data in Figure 2—figure supplement 1? What impact could this assumption have had on model results?

Please see our response above.

Reviewer #3:[…] Lastly, there should be more discussion about the risk of microcephaly. The comparisons to French Polynesia and Yap are great but there is a lot of other work that has been done, especially clinical studies: https://www.ncbi.nlm.nih.gov/pubmed/26943629 and https://www.ncbi.nlm.nih.gov/pubmed/27960197. It is especially important to understand why the estimates in this manuscript may be on the very low end of what is being reported elsewhere in studies specifically aimed at measuring that risk.

Please see our response above to the issues raised by this reviewer.

[Editors' note: the author responses to the re-review follow.]

The work and manuscript have greatly improved, and the reviewers are satisfied with the approach of the model. However, there are some remaining issues about the way aspects of the model and the results are described that need to be addressed before acceptance, as outlined below:1) The relationship between the Zika epidemic in FSA vs. the rest of Brazil is still not very clear, although it is improved from before. For example, in the last paragraph of the Introduction it states that Zika peaked elsewhere in the country in 2016. That may be true for some locations, but data suggest otherwise for many of the most affected states. As the authors now note, Zika case surveillance changed substantially in 2016. Clearly higher case numbers are associated with this, but microcephaly numbers were much higher for many states as a result of the 2015 epidemics, so in fact many of the NE states in particular likely had much bigger epidemics in 2015, and FSA may be a very good representation of what happened in those states, not an anomaly as implied in the manuscript. The authors even found evidence of a 4-fold increase in reported FSA between 2015 and 2016. In my view, this is a very important finding and emphasizes how little we know about 2015, even in area with relatively strong surveillance.While we agree that it is not very helpful to speculate about exactly when the Zika epidemic peaked in different parts of Brazil, we think the overall message is still misleading – it currently appears to suggest that the epidemic in FSA was earlier than the rest of the country when the quality of surveillance in 2015 is probably not good enough to make such conclusions. A more appropriate message could be that detailed analysis of FSA indicated a large epidemic in 2015, which could possibly have occurred in other places although this was not picked up by the limited surveillance at that time.

It was indeed our intention to speculate that FSA had a particular behaviour of peaking in 2015, when overall case counts in Brazil peaked in 2016, but we did not intend to focus on FSA being an anomaly – we understand, however, how our statements could be misleading. We also agree that relating our results of low observation rates and potential (estimated) changes of the surveillance system between 2015 and 2016 should be better contextualized for observations across the country. We have therefore followed the Editor’s advice and have now changed our statements in the main text, as highlighted below.

Changes in the Introduction section:

“The rapic accumulation of herd-immunity significantly reduced the number of cases during the following year, when total ZIKV-associated disease was peaking at the level of the country.”

Changes in the Discussion section:

The pattern of reported ZIKV infections in FSA was characterized by a large epidemic in 2015, in clear contrast to total reports at the country-lveel, peaking during 2016. Most notably for FSA was the epidemic decay in 2016 and fadeout in 2017.”

Changes in the Discussion section:

“It is also tempting to speculate that the 2015/2016 imbalance in reporting may have been a general phenomenon across Brazil. […] This, together with our conclusion that low MC risk with very high attack rates makes ZIKV a public health concern, could explain why most MC reports at the level of the country were in 2015 [de Oliveira et al., 2017], although for many regions the total reported number of ZIKV cases may have been surprisingly small that year.”

2) The addition of informative priors is sensible. It would also be helpful to have the posterior outputs so the main figures and analysis can be reproduced. This will enable groups working in other parts of Brazil (or indeed other countries) to make use of the analyses to compare to their findings. References should be given for the priors shown in Figure 2—figure supplement 2 – at the moment only the human incubation and infectious period are given priors.It would also be helpful to have a brief descriptive file that states what data are used in which figure (if not the actual plotting code itself) – e.g. it is not clear where the age distribution in Figure 1 came from.

A) We would like to note that while we completely agree that making the posteriors available is essential, we argue that the millions of data points used in the figures would make the supplementary files unnecessarily large.

As part of the previous revision, we had submitted supplementary tables (csv files) with 500 samples of the posteriors and corresponding model solutions for R0 and Re (both deterministic and stochastic in a total of 4 files). These samples are representative of the posteriors presented in both Figure 3 and 7. We apologize for not making it clear in the main text that this information had been made available, and have now changed the figure legends for Figure 3 to include reference to the supplementary files.

B)References for eta and α priors were indeed missing – we apologize for this. We have now added these references to the legend of Figure 2—figure supplement 2.

C) We had not included the age-related data in the supplementary files submitted in the last revision. We apologize for this. A new supplementary file has now been added (Dataset 2), which has been listed in the Supplementary files section and mentioned in the legend of Figure 1.

3) With regards to the actual code, we take the comment that this would run to many thousands of lines. We actually have facilities to host this on eLife, but we would be happy as an alternative if this was archived e.g. in a suitable GitHub repository.

We are totally committed for source code to be made public for transparency and research purposes (something the main author has already some background, in terms of publishing R-packages). Indeed, since submission of the last revision we have planned and offered a student project to develop an R-package with the goal of simulating the model described in this manuscript. We believe such project is a more useful way of making the model and methods described in this manuscript useful for the community and into the public domain. We would like to propose that a statement be added to the main text, on the grounds that the source code can be made available by the authors upon request.

“The approach used in this study uses code in C/C++, bash and R scripts. The code can be made available by the authors upon request.”

We note that once the R-package is developed and made public, this current manuscript submitted to *eLife* will serve as the main case study and first demonstration of its potential.

We are hopeful that the Editors and reviewers understand our position and that this does not affect the final decision from *eLife* to publish our manuscript.

4) The causal link between ZIKV infection and at least some congenital outcomes (e.g. microcephaly) is now clear from multiple studies in multiple locations. While much is left to be learned, we recommend being more direct in the second paragraph of the Introduction.

We have changed our statement to be more direct about the general consensus of a causal link:

“There is wide statistical support for a causal link between ZIKV and severe manifestations such as microcephaly [Rubin, Greene and Baden, 2016; de Araújo et al., 2016a; de Araújo et al., 2016b; Honein et al., 2017; Brasil et al., 2016; de Oliveira et al., 2017], and the proposed link in 2015 led to the declaration of the South American epidemic…”

5) The term 'herd-immunity' seems a little imprecise as immunity has unlikely reached a level that is truly protective against invasion. Perhaps 'population-level immunity' would be more appropriate?

We are unsure if this comment refers to a particular sentence or the general use of the term ‘herd-immunity’. We note that ‘herd-immunity’ is synonymous to 'population-level immunity' (and not a particular level of population-level immunity that offers protection). We have therefore applied no changes in regards to the term herd-immunity, but are happy to address this further if the editors/reviewers feel strongly about it.